# The DBL-1/TGF-β signaling pathway tailors behavioral and molecular host responses to a variety of bacteria in *Caenorhabditis elegans*

Bhoomi Madhu[1,2], Mohammed Farhan Lakdawala[1,3], Tina L Gumienny[1]*

[1]Department of Biology, Texas Woman's University, Denton, United States; [2]Perelman School of Medicine, University of Pennsylvania, Philadelphia, United States; [3]AbbVie (United States), Worcester, United States

**Abstract** Generating specific, robust protective responses to different bacteria is vital for animal survival. Here, we address the role of transforming growth factor β (TGF-β) member DBL-1 in regulating signature host defense responses in *Caenorhabditis elegans* to human opportunistic Gram-negative and Gram-positive pathogens. Canonical DBL-1 signaling is required to suppress avoidance behavior in response to Gram-negative, but not Gram-positive bacteria. We propose that in the absence of DBL-1, animals perceive some bacteria as more harmful. Animals activate DBL-1 pathway activity in response to Gram-negative bacteria and strongly repress it in response to select Gram-positive bacteria, demonstrating bacteria-responsive regulation of DBL-1 signaling. DBL-1 signaling differentially regulates expression of target innate immunity genes depending on the bacterial exposure. These findings highlight a central role for TGF-β in tailoring a suite of bacteria-specific host defenses.

*For correspondence:
tgumienny@twu.edu

Competing interest: The authors declare that no competing interests exist.

## Editor's evaluation

This study provides valuable insight into the role of the TGF-β signaling pathway in the immune response of *C. elegans*. The authors report a convincing analysis of molecular and behavioral responses to a broad panel of bacteria, dissecting the contribution of the TGF-β pathway to these responses.

## Introduction

Organisms recognize and respond to potential environmental insults by coordinating protective defenses (*Medzhitov and Janeway, 1998*). Invertebrates and vertebrates both employ conserved innate immune responses as immediate front-line protection from challenges including harmful bacteria (*Medzhitov and Janeway, 1998*; *MacGillivray and Kollmann, 2014*; *Cheesman et al., 2016*; *Pukkila-Worley, 2016*; *Sellegounder et al., 2018*). These responses are tailored to the bacterial challenge. However, how these responses are specified and what the responses are to different bacteria remains a challenge (*Akira et al., 2006*).

The roundworm *Caenorhabditis elegans* is an established model system to study the regulation of immune responses in vivo (*Engelmann and Pujol, 2010*). *C. elegans* naturally thrives in a soil environment where it feeds on bacteria and is in constant association with a diverse range of microbes that are both food and threat (*Engelmann and Pujol, 2010*; *Cheesman et al., 2016*). A limited number of microorganisms are known to infect *C. elegans*, including fungi, Gram-negative bacteria, and

Gram-positive bacteria (*Couillault and Ewbank, 2002*; *Gravato-Nobre et al., 2005*; *Begun et al., 2007*; *Wong et al., 2007*; *Singh and Aballay, 2009*; *Zugasti and Ewbank, 2009*; *Pukkila-Worley et al., 2012*; *Ahamefule et al., 2020*). *C. elegans* has an innate immune system that confers protection through behavioral, physical, and molecular mechanisms (*Engelmann and Pujol, 2010*). *C. elegans* molecular responses are tailored to the pathogen, creating 'antimicrobial fingerprints' specific to the challenge (*Alper et al., 2007*).

The molecular immune defenses in *C. elegans* induced by infection are coordinated by functionally conserved cell–cell signaling pathways including mitogen-activated protein kinase (MAPK), insulin-like, and DBL-1/TGF-β (transforming growth factor β) signaling pathways (*Gravato-Nobre et al., 2005*; *Begun et al., 2007*; *Wong et al., 2007*; *Singh and Aballay, 2009*; *Alper et al., 2007*; *Zugasti and Ewbank, 2009*; *Berg et al., 2019*). These pathways regulate an overlapping set of target defense genes, indicative of coordinated crosstalk between these signaling pathways. The roles of MAPK and insulin-like signaling pathways in immune responses to a wide variety of bacteria are well characterized. While the role of DBL-1 in defending nematodes from a few strongly pathogenic Gram-negative bacteria is reported, its role in protection against Gram-positive bacteria has not been well characterized (*Tan et al., 1999*; *Mallo et al., 2002*; *Zhang and Zhang, 2012*). Previous reports indicate that the DBL-1 pathway affects expression of many immune response-associated genes including lectins, saposin-like proteins, and lysozymes (*Mochii et al., 1999*; *Wong et al., 2007*; *Liang et al., 2007*; *Roberts et al., 2010*). However, a role for DBL-1/TGF-β signaling in eliciting robust targeted immune responses to different bacterial challenges is not well defined in *C. elegans* or other organisms (*Kim et al., 2002*; *Mallo et al., 2002*; *Murphy et al., 2003*; *Garsin et al., 2003*; *Alper et al., 2007*).

In this work, we dissected the roles of DBL-1/TGF-β signaling in regulating a broad array of microbe-specific defense responses in *C. elegans*. Using behavioral and molecular approaches, we identified DBL-1-dependent and -independent defense responses that are tailored to the specific bacterial exposure. In addition, we showed that DBL-1 signaling is induced in response to Gram-negative bacteria but is repressed in response to Gram-positive bacteria. We propose that animals lacking DBL-1 signaling respond with heightened avoidance behaviors and reduced feeding to selected bacterial environments because they perceive the environment as more hostile. Collectively, our findings highlight a central role for DBL-1 in regulating a suite of bacteria-specific host defenses and also demonstrate bacteria-responsive regulation of DBL-1 signaling.

## Results
### Loss of DBL-1 reduces lifespan of animals fed on specific bacteria

To study the role of DBL-1 in specific host responses to bacteria at behavioral, molecular, and physiological levels, we first established a panel of bacteria that would facilitate genetic and molecular studies over time: previous studies have been limited in range of challenge and used pathogens that killed animals in hours or a few days. The control bacteria chosen was Gram-negative *Escherichia coli* OP50, a commonly used strain for laboratory culture of *C. elegans*. The panel of test bacteria comprised Gram-negative and Gram-positive bacteria that are opportunistic pathogens in humans and are found in the natural habitat of *C. elegans* (*Samuel et al., 2016*). We selected three Gram-negative test strains, *Serratia marcescens* (D1), *Enterobacter cloacae* (ATCC 49141), and *Klebsiella oxytoca* (ATCC 49131), and three Gram-positive test strains, *Bacillus megaterium* (ATCC 14581), *Enterococcus faecalis* (ATCC 51299), and *Staphylococcus epidermidis* (ATCC 49134).

We first asked if these bacteria alter lifespan of wild-type *C. elegans*. While pathogenic bacteria can reduce *C. elegans* lifespan, other microbial diets can extend lifespan of *C. elegans* populations (*Stuhr and Curran, 2020*). We noted an extended lifespan of wild-type animals on two Gram-negative bacteria (*E. cloacae* and *K. oxytoca*, by 4 d and 2 d, respectively). However, we found that wild-type animals on Gram-negative *S. marcescens* have lifespans comparable to *E. coli*-fed animals. Interestingly, we observed an extended lifespan of wild-type animals on all three Gram-positive bacteria (*B. megaterium*, *E. faecalis*, and *S. epidermidis*) (*Supplementary file 1*).

Loss of DBL-1 has previously been shown to reduce lifespan of animals exposed to fungus *Drechmeria coniospora*, Gram-negative strains *E. coli* and *S. marcescens* Db11, and Gram-positive *E. faecalis* (*Mallo et al., 2002*; *Zugasti and Ewbank, 2009*; *Tenor and Aballay, 2008*). The reduced lifespan of animals lacking DBL-1 on *E. coli* OP50 was not observed when animals were grown on

5-fluorodeoxyuridine (FUdR), which blocks DNA synthesis of both bacteria and the nematodes (*Mallo et al., 2002*). It has also been shown that DBL-1 is required for the probiotic *Lactobacillus* spp. Lb21-mediated lifespan extension of *C. elegans* subjected to pathogenic methicillin-resistant *Staphylococcus aureus* (MRSA) (*Mørch et al., 2021*). To determine if DBL-1 is required in maintaining lifespan of animals subjected to our bacterial panel, we compared lifespans of *dbl-1(-)* and wild-type animals exposed to the control or test bacteria on plates containing FUdR. Loss of DBL-1 did not alter lifespan of animals fed on plates containing *E. coli* OP50 and FUdR (*Figure 1A*), consistent with previous work (*Mallo et al., 2002*). *dbl-1* mutant animals did not have the lifespan extension seen in wild-type populations on *E. cloacae* or *K. oxytoca*: lifespans of *dbl-1* mutant populations were the same on these two Gram-negative bacteria as on *E. coli* (*Figure 1B and C*, *Supplementary file 1*). However, loss of DBL-1 resulted in a significantly shortened lifespan on *S. marcescens* compared to *E. coli* control bacteria, unlike the wild-type (by 6 d, *Supplementary file 1*). This reduced lifespan of *dbl-1(-)* animals was significantly lower than that of wild-type animals fed on *S. marcescens*, which is consistent with a previous report that used the more virulent *S. marcescens* strain Db11 (*Figure 1D*; *Mallo et al., 2002*). The lifespan of *dbl-1(-)* animals was modestly extended upon exposure to Gram-positive *B. megaterium* and *E. faecalis* compared to the *E. coli*-fed population's lifespan, but was significantly less extended than the wild-type lifespan (by about a day, *Figure 1E and F*, *Supplementary file 1*). Lastly, *dbl-1* mutant animals displayed a significantly decreased lifespan on *S. epidermidis* compared to *E. coli*-fed populations' lifespan (by almost 2 d), which was not observed in wild-type animals fed on *S. epidermidis* (*Figure 1G*, *Supplementary file 1*).

In summary, we showed that exposure to all bacteria in the test panel except *S. marcescens* resulted in lifespan extension of wild-type animals. Loss of DBL-1 reduced lifespan in all test bacteria in this panel to varying degrees. *dbl-1(-)* mutant animals have the lowest average lifespan on *S. marcescens*, significantly lower than the same strain on control *E. coli* OP50. In contrast to the wild type, *dbl-1(-)* animals fed *S. epidermidis* have reduced lifespans compared to the same strain on control *E. coli* OP50. Wild-type lifespan extension is lost in *dbl-1(-)* mutant animals fed *E. cloacae* and *K. oxytoca*, but their average lifespan is comparable to the same strain grown on *E. coli* OP50. On *B. megaterium* and *E. faecalis*, the lifespan of *dbl-1(-)* mutants was extended but to a lesser extent than the wild type (*Figure 1B–G*). Together, these results provide evidence that DBL-1 signaling confers protection against some bacteria and contributes to the lifespan extension observed in response to most bacteria in our panel.

## DBL-1 does not alter intestinal integrity of animals upon exposure to specific bacteria

One cause of reduced survival upon exposure to pathogenic bacteria is disruption of intestinal integrity (*Sim and Hibberd, 2016*). To examine if this is the underlying reason for DBL-1-dependent differences in animal survival in response to various bacteria, we investigated the integrity of the intestinal barrier of animals fed on different bacteria over time. We compared the intestinal barrier function of wild-type and *dbl-1(-)* animals exposed to control and test bacteria when 50% of the population on the test bacteria was still alive using a cell-impermeable blue dye that remains in the intestinal lumen in healthy animals (*Gelino et al., 2016*). Animals with intact intestinal barrier function will contain dye only within the intestinal lumen, but animals with intestinal barrier dysfunction will have leakage of the dye from the intestine into the body cavity. Consistent with previous reports, wild-type populations on *E. coli* exhibited an increased percentage of animals with dye outside the intestine over time, indicating an age-dependent reduction of intestinal integrity (*Gelino et al., 2016*). *dbl-1*-mutant populations grown on *E. coli* also displayed a decline in intestinal integrity that was not significantly different from the decline observed in the wild-type populations at the tested time points (*Figure 1—figure supplement 1A*). Exposure of wild-type animals to any of the test bacteria in the panel did not further decrease intestinal integrity compared to *E. coli* (*Figure 1—figure supplement 1*). Similar to wild-type animals, exposure of *dbl-1(-)* mutants to the test bacteria did not further significantly decrease their intestinal integrity. Therefore, DBL-1 is not required for intestinal integrity or its age-related decline. Furthermore, this result suggests the lifespan changes observed in *dbl-1* mutant populations are not caused by loss of intestinal integrity.

Lifespan can also be reduced by increased bacterial colonization in the *C. elegans* gut (*Portal-Celhay et al., 2012*). To determine if the decrease in lifespan of *dbl-1(-)* animals is correlated with

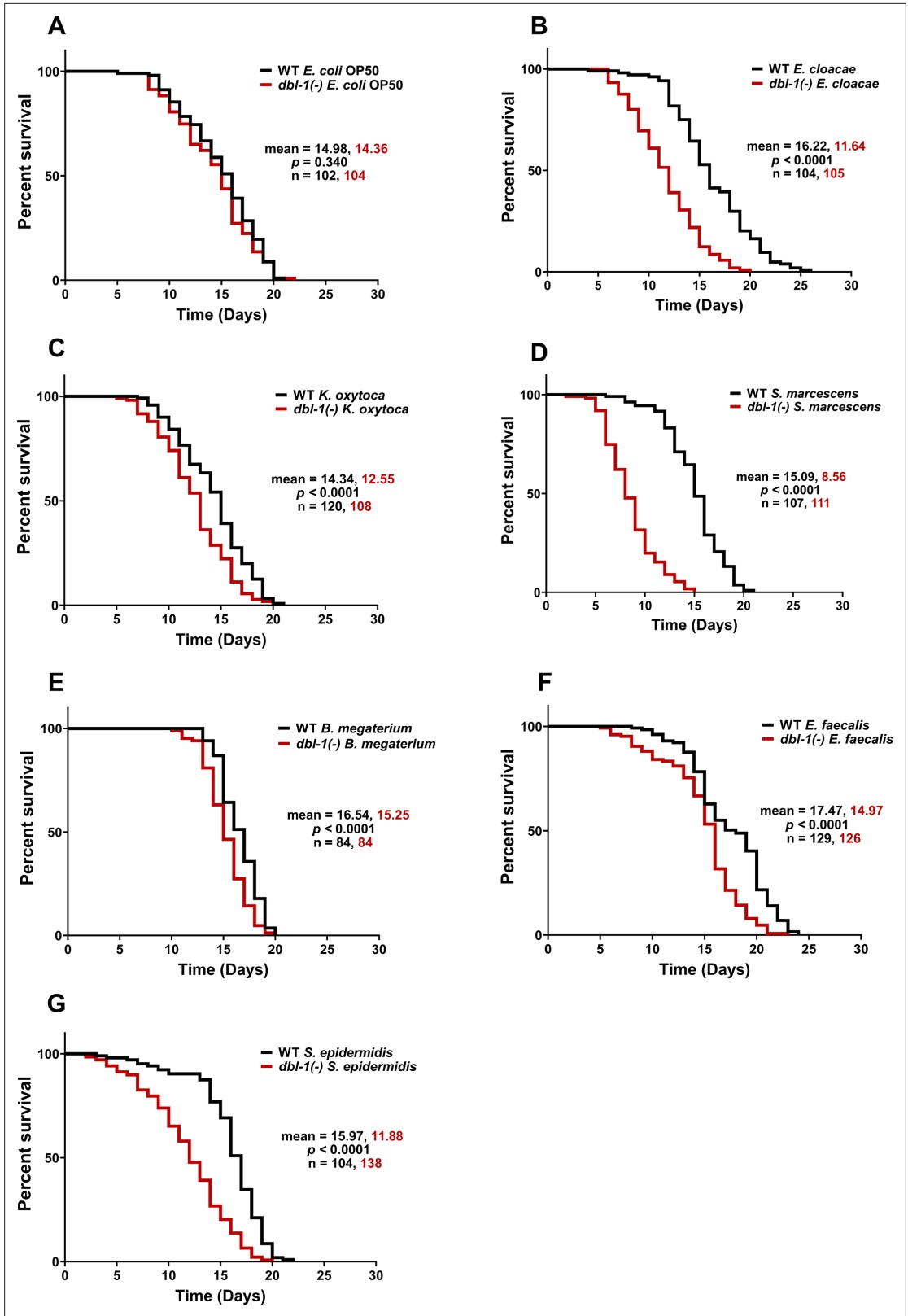

**Figure 1.** Loss of DBL-1 decreases lifespan of animals exposed to test Gram-negative and Gram-positive bacteria. Wild-type and *dbl-1(-)* animals were scored for survival over time from the L4 stage (t = 0 hr) on the following bacteria: (**A**) *E. coli* OP50 (control), (**B**) *E. cloacae*, (**C**) *K. oxytoca*, (**D**) *S. marcescens*, (**E**) *B. megaterium*, (**F**) *E. faecalis*, and (**G**) *S. epidermidis*. Survival fraction was calculated by the Kaplan–Meier method. p-Values were

*Figure 1 continued on next page*

*Figure 1 continued*

calculated using log-rank test and p<0.01 compared to wild-type animals exposed to the same bacteria was considered significant. One representative trial is presented.

The online version of this article includes the following source data and figure supplement(s) for figure 1:

**Source data 1.** Related to *Figure 1*.

**Figure supplement 1.** Loss of DBL-1 does not affect intestinal integrity upon exposure to specific bacteria.

**Figure supplement 1—source data 1.** Related to *Figure 1—figure supplement 1*.

**Figure supplement 2.** Loss of DBL-1 does not affect bacterial colonization upon exposure to *S. marcescens*.

**Figure supplement 2—source data 1.** Related to *Figure 1—figure supplement 2*.

an increase in gut colonization, we quantified bacterial load in the intestine of wild-type and *dbl-1(-)* animals fed *S. marcescens*, the bacterial strain in the selected panel that had the most deleterious lifespan effect on *dbl-1(-)* populations. No significant differences in bacterial load were observed between wild-type and *dbl-1(-)* animals that were exposed to *S. marcescens* (*Figure 1—figure supplement 2*). This suggests that bacterial colonization is not the underlying cause for reduction in *dbl-1(-)* populations' survival upon exposure to *S. marcescens*.

## Loss of DBL-1 and exposure to specific bacteria reduce feeding

We next asked if our panel of bacteria affect feeding behavior in *C. elegans*. *C. elegans* reduce their food intake in response to bacteria they perceive as harmful (*O'Quinn et al., 2001*). To determine if loss of DBL-1 alters animals' perception of test bacteria as potential threat, we assessed pharyngeal pumping of wild-type and *dbl-1(-)* populations fed on control and test bacteria. Loss of DBL-1 does not alter the pharyngeal pumping rate of animals fed on the control *E. coli* OP50 bacteria (*Figure 2A and B*). Overall, animals fed the Gram-negative bacteria did not have major changes in pharyngeal

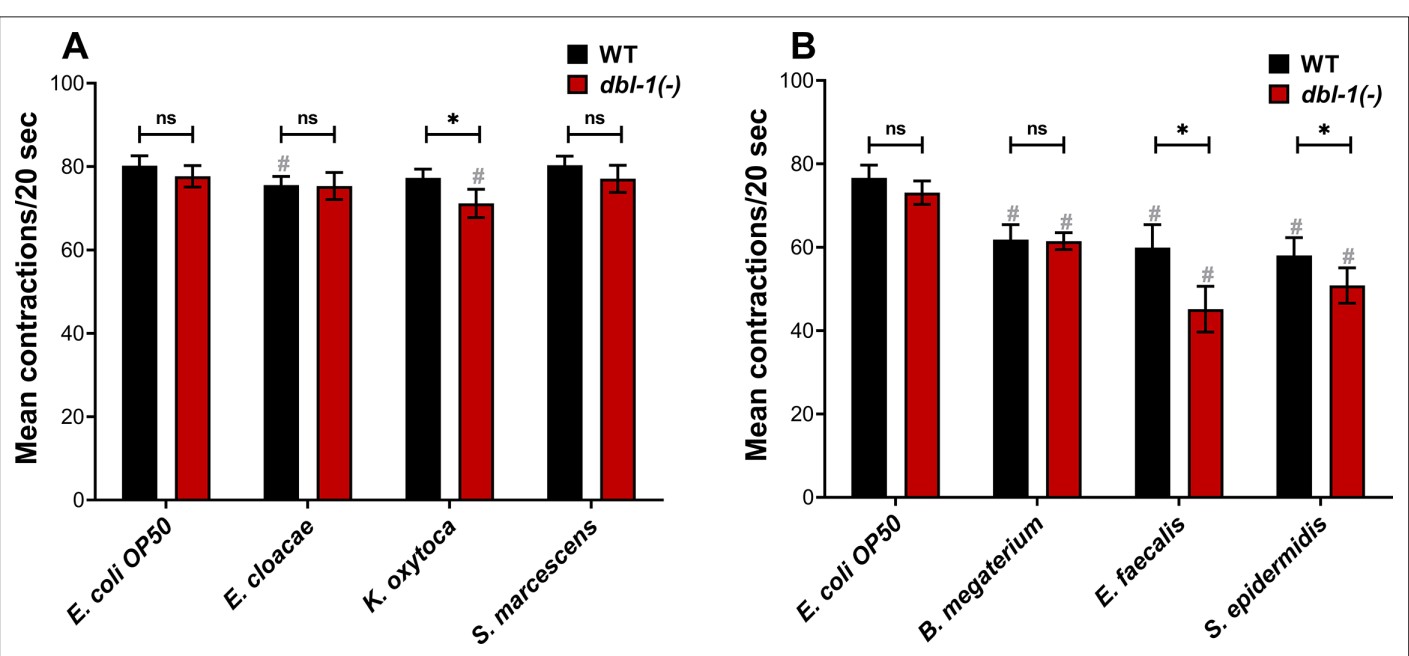

**Figure 2.** Loss of DBL-1 and exposure to specific bacteria results in decreased pharyngeal pumping. Wild-type and *dbl-1(-)* animals at the L4 stage were exposed to the following bacteria: (**A, B**) *E. coli* OP50 (control); (**A**) *E. cloacae, K. oxytoca, S. marcescens*; (**B**) *B. megaterium, E. faecalis*, or *S. epidermidis*. After 48 hr of exposure, the number of pharyngeal pumps was counted twice per 20 s. The pharyngeal pumps were averaged for each animal. One representative trial is presented. Error bars represent standard deviation. n = 8–12 per condition. *p<0.01, ns, not significant, compared to wild-type animals exposed to the same bacteria, and #p<0.01, respective genotype exposed to test bacteria in comparison to control bacteria by two-way ANOVA using Tukey's post hoc test.

The online version of this article includes the following source data for figure 2:

**Source data 1.** Related to *Figure 2*.

pumping rates (*Figure 2A*). Wild-type animals fed on *E. cloacae* exhibited a small but significant decrease in pharyngeal pumping that was not further decreased in *dbl-1(-)* animals. Pharyngeal pumping of wild-type animals was not significantly reduced after exposure to *K. oxytoca*, but animals lacking DBL-1 had a mild but significant decrease in pumping rate. Wild-type and *dbl-1(-)* animals fed on *S. marcescens* had a similar pumping rate as on the control bacteria (*Figure 2A*). There is no clear correlation between lifespan phenotypes and feeding rate for populations fed on the Gram-negative bacteria. For wild-type animals fed on the three Gram-positive bacteria, though, the pharyngeal pumping rate was dramatically decreased, consistent with the lifespan extension these strains conferred to wild-type *C. elegans* (*Figure 2B*, *Supplementary file 1*). Interestingly, the feeding rate of *dbl-1(-)* animals is further reduced from the wild-type rate on *E. faecalis* and *S. epidermidis*. However, there is no reproducibly significant decrease in the pharyngeal pumping rate of *dbl-1(-)* animals fed on *B. megaterium* in comparison to that of the wild type (*Figure 2B*). These results indicate that while the feeding reduction caused by exposure to the Gram-positive bacteria is partially independent of DBL-1, a stronger pharyngeal pumping reduction in *dbl-1(-)* populations occurs in response to some bacteria, providing support to the idea that loss of DBL-1 sensitizes animals to certain bacterial stressors.

## DBL-1 signaling is required to suppress avoidance of specific Gram-negative but not Gram-positive bacteria

*C. elegans* can also respond to potentially pathogenic bacteria or harmful environments by avoiding such challenges (*Pradel et al., 2007*; *Beale et al., 2006*; *Anderson and McMullan, 2018*). To further investigate the role of DBL-1 in perceiving harmful environments, we performed an avoidance assay. First, we measured the avoidance response of wild-type animals fed on the test bacteria over the first 2 d of adulthood and compared it with the avoidance response on the control bacteria. Wild-type animals do not avoid *E. coli* (*Figure 3A*). We found that wild-type animals do not avoid *E. cloacae* but do avoid Gram-negative *K. oxytoca* and *S. marcescens* (*Figure 3B–D*). While wild-type animals displayed a moderate to strong response to Gram-positive *B. megaterium*, they exhibited mild or no avoidance of *E. faecalis* and *S. epidermidis* (*Figure 3E–G*). This indicates that wild-type animals perceive *K. oxytoca*, *S. marcescens*, and *B. megaterium* as harmful.

Next, we tested avoidance behavior of *dbl-1(-)* populations exposed to the test bacteria and compared it with wild-type animals fed on the same test bacteria. Loss of DBL-1 did not result in an avoidance response to the control bacteria (*Figure 3A*). Loss of DBL-1 resulted in a significantly stronger avoidance response to all three tested Gram-negative bacteria (*Figure 3B–D*). Interestingly, the avoidance response of *dbl-1(-)* animals upon exposure to Gram-positive *B. megaterium*, *E. faecalis*, and *S. epidermidis* was similar to wild type (*Figure 3E–G*). The protective behavioral response displayed by *dbl-1(-)* populations on bacteria that wild-type animals do not avoid suggests that animals lacking DBL-1 perceive the tested Gram-negative (but not Gram-positive bacteria) to be more harmful than wild-type animals do. This also suggests that DBL-1 signaling normally suppresses avoidance depending on the specific bacterial exposure.

## Canonical DBL-1 signaling suppresses avoidance to Gram-negative bacteria but is dispensable for avoidance to Gram-positive bacteria

Because we observed strong bacteria-specific avoidance responses in animals lacking DBL-1, we next asked if the canonical DBL-1 signaling pathway is required to attenuate this response. Canonical signaling occurs by DBL-1 ligand binding to receptors SMA-6 and DAF-4, which activate downstream Smad transcription factors SMA-2, SMA-3, and SMA-4 (*Savage et al., 1996*). A non-canonical DBL-1 pathway, which does not signal through SMA-2 and SMA-4, is required for *C. elegans* to respond to the fungus *D. coniospora* (*Zugasti and Ewbank, 2009*). To identify the Smads required for DBL-1 pathway-mediated bacterial avoidance, we measured the avoidance response of *sma-2(-)*, *sma-3(-)*, and *sma-4(-)* strains fed on the test Gram-negative or Gram-positive bacteria and compared them with the avoidance response on the control bacteria. On the control bacteria, loss of *sma-2* and *sma-3* did not result in significantly increased avoidance. However, populations with *sma-4(e727)* showed a moderate but significant avoidance response to the control bacteria (*Figure 3A*). Because *sma-4(e727)*, a strong loss-of-function point mutation (Q246Stop), initially displayed a stronger avoidance response, we confirmed this effect was reproducible in the *sma-4(jj278)* null strain, in which most of

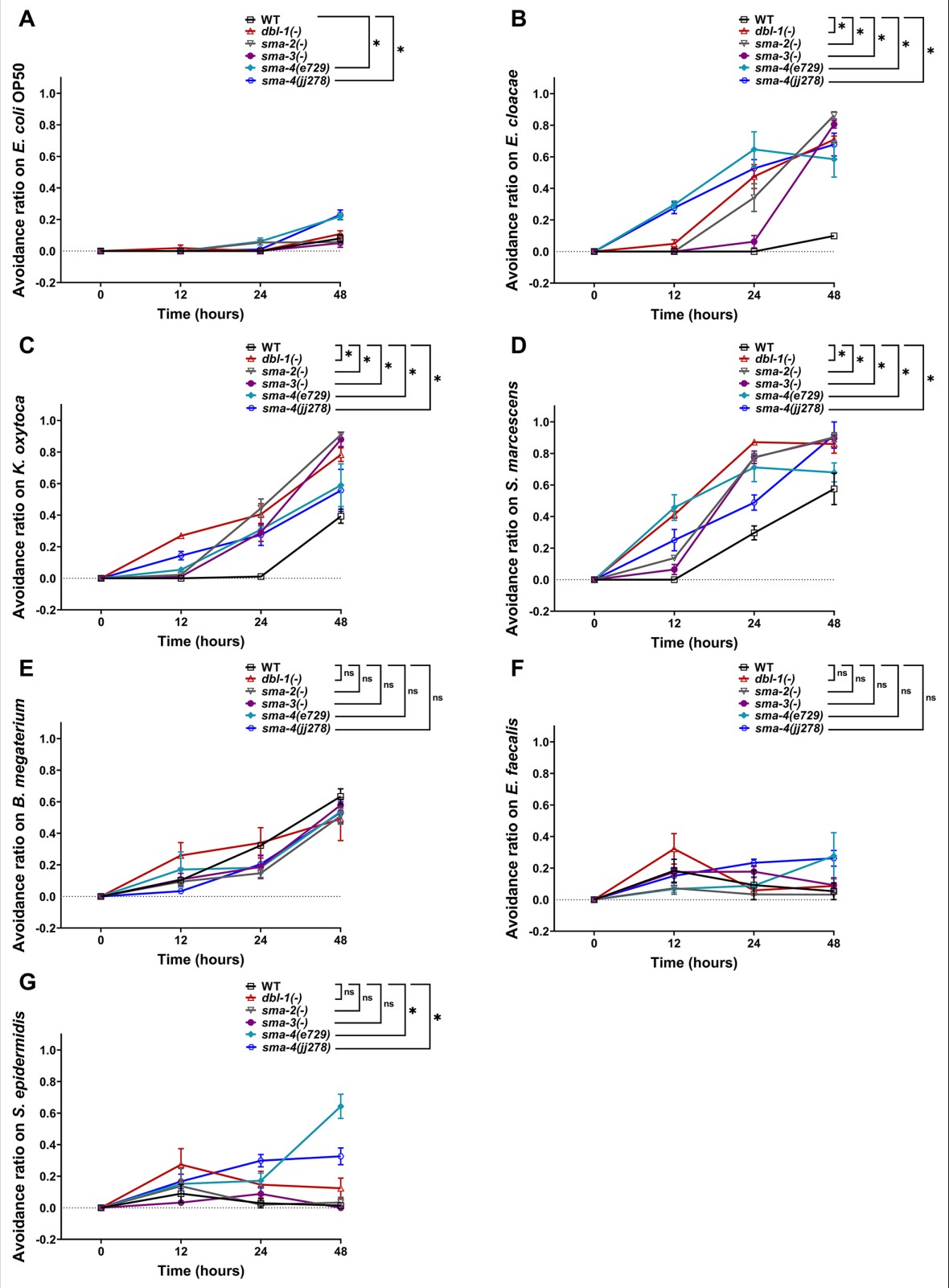

**Figure 3.** Avoidance to Gram-negative bacteria increases upon loss of canonical DBL-1 signaling over time while avoidance to specific Gram-positive is independent of DBL-1 signaling. Wild-type, *dbl-1(-)*, *sma-2(-)*, *sma-3(-)*, and two *sma-4(-)* strains at the L4 stage (t = 0 hr) were exposed to the following bacteria: (**A**) *E. coli* OP50 (control), (**B**) *E. cloacae*, (**C**) *K. oxytoca*, (**D**) *S. marcescens*, (**E**) *B. megaterium*, (**F**) *E. faecalis*, or (**G**) *S. epidermidis*. The avoidance ratio was calculated and compared to the wild-type control animals. Each trial used three plates of 30 animals each per condition. One

*Figure 3 continued on next page*

*Figure 3 continued*

representative trial is presented. Error bars represent standard error mean. *p<0.05, ns, not significant, compared to wild-type animals exposed to the same bacteria by repeated measures ANOVA using Tukey's post hoc test.

The online version of this article includes the following source data for figure 3:

**Source data 1.** Related to *Figure 3*.

the open reading frame was deleted using CRISPR (*McKillop et al., 2018*). *sma-2(-)*, *sma-3(-)*, and both *sma-4(-)* populations displayed strong, reproducible avoidance responses to all Gram-negative bacterial strains tested that were comparable to the *dbl-1(-)* populations' response (*Figure 3B–D*).

In comparison, the response of Smad mutant populations to the panel of Gram-positive bacteria was notably different. Loss of *sma-2*, *sma-3*, or *sma-4* resulted in avoidance responses on the Gram-positive *B. megaterium* and *E. faecalis* similar to the wild type and *dbl-1(-)* responses (*Figure 3E and F*). This indicates that avoidance elicited in response to these bacterial strains is independent of not only DBL-1, but also of the Smad machinery. However, both *sma-4* mutant strains but not *dbl-1*, *sma-2*, or *sma-3* strains had a stronger, moderate avoidance response on the Gram-positive *S. epidermidis* in comparison to the wild type (*Figure 3G*). Together, our findings indicate that while canonical DBL-1 signaling does not play a major role in responding to *E. coli*, a standard Gram-negative food source, and Gram-positive bacteria, it plays a major role in suppressing animal avoidance responses to the tested Gram-negative bacteria.

### *sma-4* expression is specifically induced in response to Gram-positive bacteria

We next asked if animals alter expression of the DBL-1 Smads in response to the panel of opportunistic pathogens. We tested gene expression levels of *sma-2*, *sma-3*, and *sma-4* in wild-type and *dbl-1* mutant backgrounds in response to our panel of bacteria. Gene expression was assessed by real-time PCR experiments that were performed in triplicate in three independent trials. Expression of mRNA from these three Smad genes was not changed in *dbl-1(-)* animals on control *E. coli* OP50 compared to the wild type (*Figure 4—figure supplement 1A*). The relative levels of *sma-2* mRNA were consistently decreased in the *dbl-1* mutant background upon exposure to all test bacterial strains except *S. epidermidis*, where *sma-2* mRNA levels were increased in both the wild-type and *dbl-1(-)* backgrounds (*Figure 4A and B* and *Figure 4—figure supplement 1A and B*). The relative levels of *sma-3* mRNA were not reproducibly different between *dbl-1(-)* and wild-type backgrounds exposed to the same control or test bacteria (*Figure 4—figure supplement 1C and D*). Whereas *sma-3* expression levels remained largely unchanged in wild-type animals when exposed to Gram-negative and Gram-positive bacteria, exposure to *E. faecalis* or *S. epidermidis* resulted in increased expression of *sma-3* in the *dbl-1(-)* background compared to control bacteria (*Figure 4C and D*). On test Gram-negative bacteria, *sma-4* expression was similar in the *dbl-1(-)* background as in the wild type (*Figure 4E*, *Figure 4—figure supplement 1E*). However, *sma-4* was significantly induced in response to Gram-positive bacteria in both wild-type and *dbl-1* mutant backgrounds, albeit less in the *dbl-1(-)* populations (*Figure 4F* and *Figure 4—figure supplement 1F*). Together, these results suggest that the DBL-1 Smads are differently regulated at the level of gene expression by molecular pathways that are responsive to specific bacterial challenges. *sma-2*, but not *sma-3*, requires DBL-1 for full expression on all bacterial food sources except *S. epidermidis*. Neither *sma-2* nor *sma-3* expression is affected to a great extent by test bacteria. In contrast, *sma-4* expression is not altered by Gram-negative bacteria, but is responsive to Gram-positive bacteria in the panel, possibly by both DBL-1 and DBL-1-independent mechanisms. The Smads may be regulated post-transcriptionally to contribute to the differential avoidance responses to various bacteria. Investigation of downstream genes regulated by Smads will further our understanding of the role of Smads in differential avoidance responses.

### DBL-1 signaling is activated in response to specific Gram-negative bacteria and is repressed in response to Gram-positive bacteria

We next asked if DBL-1 signaling activity is altered in response to the bacterial panel. To address this question, we fed the bacterial panel to wild-type or *dbl-1(-)* animals expressing an integrated fluorescent DBL-1 pathway reporter called RAD-SMAD. This reporter consists of the GFP gene under the

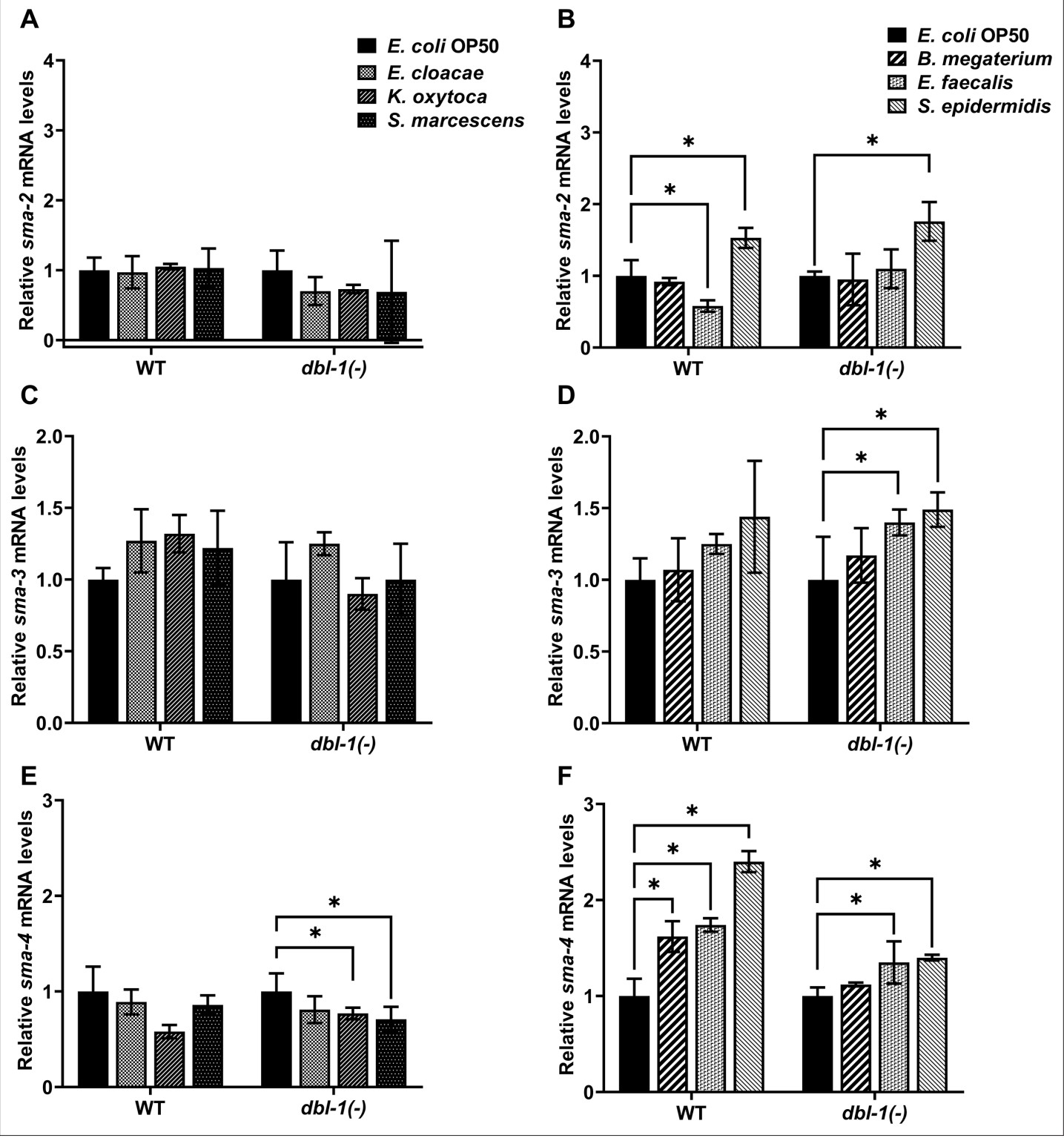

**Figure 4.** Smad transcription factors gene expression is altered by specific bacteria. Wild-type and *dbl-1(-)* animals at the L4 stage were exposed to *E. coli* OP50 (control), *E. cloacae*, *K. oxytoca*, *S. marcescens*, *B. megaterium*, *E. faecalis*, or *S. epidermidis* for 48 hr. Relative mRNA expression levels of (**A, B**) *sma-2*, (**C, D**) *sma-3*, and (**E, F**) *sma-4* were quantitated by real-time PCR. Experiments were performed in three technical replicates and in three independent biological trials. One representative trial is presented. Error bars represent standard deviation. *p<0.05, mRNA expression level in respective genotype exposed to test bacteria compared to control *E. coli* OP50, by one-way ANOVA with Dunnett's multiple-comparisons test.

The online version of this article includes the following source data and figure supplement(s) for figure 4:

*Figure 4 continued on next page*

*Figure 4 continued*

**Source data 1.** Related to *Figure 4*.

**Figure supplement 1.** Smad transcription factor gene expression is altered by specific bacteria.

**Figure supplement 1—source data 1.** Related to *Figure 4—figure supplement 1*.

control of multiple copies of a Smad-binding element sequence (*Tian et al., 2010*). The expression of this reporter is robust in hypodermal nuclei of L2 stage wild-type animals, and changes in DBL-1 signaling positively correlate with hypodermal expression of RAD-SMAD (*Tian et al., 2010*; *Fernando et al., 2011*; *Gleason et al., 2014*; *Liu et al., 2015*; *Savage-Dunn et al., 2019*). We measured the hypodermal fluorescence intensity of the RAD-SMAD-expressing L2 populations fed control or test Gram-negative bacteria. Compared to the wild-type expression of RAD-SMAD fluorescence on *E. coli* control bacterial conditions (*Figure 5A*), reporter fluorescence in the wild-type background on Gram-negative bacteria *E. cloacae* and *K. oxytoca* was strikingly increased (*Figure 5B, C, and H*). While Gram-negative *S. marcescens* increased expression of the DBL-1 reporter in the population, it did not reach significance (p=0.49, *Figure 5D and H*).

In stark contrast, RAD-SMAD hypodermal fluorescence in wild-type animals was lost upon exposure to test Gram-positive bacteria. Because of the general absence of measurable hypodermal fluorescence, we compared the fraction of wild-type RAD-SMAD population exhibiting detectable GFP fluorescence when exposed to control or test Gram-positive bacteria. RAD-SMAD hypodermal fluorescence in wild-type populations was significantly reduced upon exposure to Gram-positive *B. megaterium* (0% had any detectable expression [n = 44]) or *S. epidermidis* (6% [n=66]) (*Figure 5E, G, and I*). RAD-SMAD hypodermal fluorescence was detected only in half the wild-type population fed *E. faecalis* (50% [n=52]), and fluorescence levels were much fainter in those animals with detectable RAD-SMAD in the hypoderm (*Figure 5F and I*, *Supplementary file 2*).

We then asked if the changes in the DBL-1 pathway reporter in response to the bacterial panel were dependent on the DBL-1 ligand. In *dbl-1(-)* L2 animals fed on *E. coli*, RAD-SMAD fluorescence in epidermal nuclei was not detectable or very faint, consistent with published reports (*Supplementary file 2*). Reporter intensity in *dbl-1(-)* animals on all Gram-negative bacteria remained mostly undetectable or faint, though a few animals surprisingly had wild-type fluorescence levels (*Supplementary file 2*). For animals lacking DBL-1 fed on any of the Gram-positive bacterial conditions, hypodermal RAD-SMAD fluorescence was undetectable or very faint. In general, these results indicate that animals modulate DBL-1 signaling depending on the specific bacterial challenge that animals encounter. In our panel, animals increased DBL-1 signaling in response to specific Gram-negative bacteria but repressed DBL-1 signaling in response to the tested Gram-positive bacteria. Of note, the increased expression of the RAD-SMAD reporter in some DBL-1-deficient animals on Gram-negative bacteria suggests that DBL-1 pathway signaling can be stimulated by pathogen response downstream of the DBL-1 ligand.

## DBL-1 mediates both common and specific gene expression responses to Gram-negative and Gram-positive bacteria

We then asked if this differential modulation of DBL-1 signaling activity translated to bacterial-specific downstream transcriptional responses. To identify the role of DBL-1 in differentially regulating transcription of downstream genes, we performed RNA sequencing using wild-type and *dbl-1(-)* animals exposed to control *E. coli* OP50, Gram-negative *S. marcescens*, or Gram-positive *E. faecalis*, building on established work that shows these bacteria alter innate immunity gene expression responses in wild-type *C. elegans* (*Mallo et al., 2002*; *Alper et al., 2007*; *Liang et al., 2007*; *Roberts et al., 2010*; *Lakdawala et al., 2019*). Animals were hatched and grown on *E. coli* OP50, synchronized as L4s, and then fed on the control or test bacteria for 48 hr before analysis. In animals lacking DBL-1 and fed control bacteria, 83 genes were significantly downregulated and 49 genes were significantly upregulated compared to the wild type (p<0.01, *Figure 6A*). Using WormCat, we identified enrichment of differentially expressed genes in stress response and extracellular material categories (*Figure 6A*). Some genes and gene classes previously reported to be regulated by DBL-1 at different developmental stages were also regulated by DBL-1 in the 2-day adult populations in this study (*Figure 6— source data 1*; *Mallo et al., 2002*; *Liang et al., 2007*; *Roberts et al., 2010*; *Lakdawala et al., 2019*).

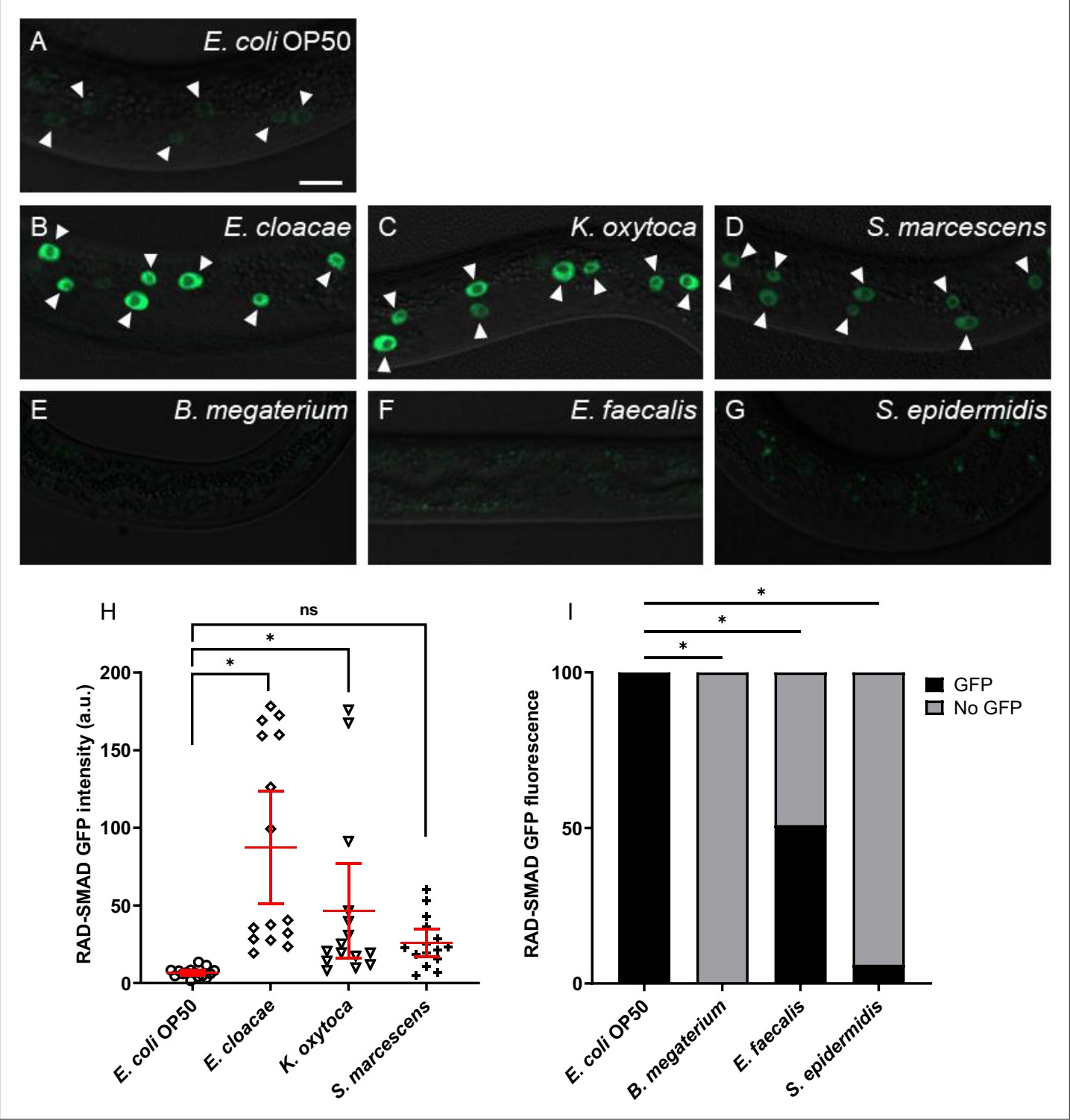

**Figure 5.** DBL-1 signaling is activated upon exposure to Gram-negative bacteria but is repressed in response to Gram-positive bacteria. Representative images of L2-stage wild-type animals expressing the RAD-SMAD reporter exposed to (**A**) *E. coli* OP50 (control), (**B**) *E. cloacae*, (**C**) *K. oxytoca*, (**D**) *S. marcescens*, (**E**) *B. megaterium*, (**F**) *E. faecalis*, or (**G**) *S. epidermidis*. Imaging conditions were consistent in (**A–G**). Arrowheads indicate visibly fluorescent hypodermal nuclei. Scale bar, 10 µm. Fluorescent intensities of wild-type animals expressing the RAD-SMAD reporter and fed on control *E. coli* OP50 or test Gram-negative bacteria are compared in (**H**). Mean RAD-SMAD fluorescence intensity of five hypodermal nuclei per animal was determined and compared. n = 10 animals per condition in each trial. Experiments were performed in three independent trials. One representative trial is presented. Error bars represent 95% confidence intervals. (**I**) Qualitative assessments of fluorescence from wild-type animals expressing the RAD-SMAD reporter

*Figure 5 continued on next page*

*Figure 5 continued*

on control *E. coli* OP50 or test Gram-positive bacteria. Percentage of animals showing detectable or no detectable RAD-SMAD GFP fluorescence from three trials is presented (n=at least 10 animals/trial). *p<0.05, mean fluorescence intensity in wild-type background on test bacteria compared to control bacteria by chi-square test.

The online version of this article includes the following source data for figure 5:

**Source data 1.** Related to *Figure 5*.

In *dbl-1(-)* animals fed on Gram-negative *S. marcescens*, 102 genes were downregulated and 117 genes were upregulated. Gene enrichment analysis revealed that these genes belonged to genes regulated by multiple stresses, CUB-like domain genes, and C-type lectins (*Figure 6B*). Almost half of all genes that were identified as highly expressed in wild-type populations in response to *S. marcescens* were also identified in the *dbl-1(-)* populations, suggesting that DBL-1 plays a lesser role in the organismal response to *S. marcescens* (*Figure 6—figure supplement 1A–C*).

In *dbl-1(-)* animals fed on Gram-positive *E. faecalis*, 63 genes were downregulated and 64 genes were upregulated compared to the wild type. These genes were assigned to the functional category of genes regulated by multiple stresses upon analysis using WormCat (*Figure 6C*). This lower number of highly regulated genes in *dbl-1(-)* animals fed on *E. faecalis* is consistent with the reduced DBL-1 reporter fluorescence—and therefore decreased DBL-1 pathway signaling—in wild-type animals on *E. faecalis* (*Figures 5 and 6C*). While there was some overlap in the genes regulated in response to *E. faecalis* independent of DBL-1, we observed a higher number of genes differentially regulated in response to *E. faecalis* exposure in wild-type populations (*Figure 6—figure supplement 1D–F*).

Notably, some highly regulated genes in the wild-type strain were common in response to both pathogenic bacterial strains, but many more were specific to the bacterial challenge (*Figure 6—figure supplement 1G*). In the *dbl-1(-)* strain, there were also genes that were differentially regulated compared to the control that were either common to both *S. marcescens* and *E. faecalis* responses or unique to one or the other challenge (*Figure 6—figure supplement 1H*). Using WormCat, gene enrichment analysis of genes regulated in response to *S. marcescens* or *E. faecalis* exposure revealed differential regulation of sets of genes associated with pathogen and stress responses, among other gene classes (*Figure 6—figure supplement 1A, B, D, and E*; *Holdorf et al., 2020*). Comparison of the differentially expressed genes between wild-type and *dbl-1(-)* strains showed many common genes regulated in response to *S. marcescens*, indicating that their regulation is independent of DBL-1 signaling (*Figure 6—figure supplement 1C*). On the other hand, there is little overlap between the *E. faecalis*-enriched genes in the wild-type and *dbl-1(-)* strains, indicating the uniqueness of genes regulated in a DBL-1-dependent and independent manner (*Figure 6—figure supplement 1F*). These results suggest that animals use DBL-1 signaling to tailor transcriptional responses specific to the type of bacterial exposure.

We focused on the DBL-1-regulated genes that have known or putative roles in innate immunity (*Figure 6D–F*). Some genes were induced in wild-type animals upon exposure to *S. marcescens* or *E. faecalis*, and this induction was lost in *dbl-1(-)* animals. We also found some genes to be upregulated only upon loss of DBL-1 and not in the wild-type animals in response to *S. marcescens* or *E. faecalis* (*Figure 6—figure supplement 1A–F*). Some gene classes are highly regulated in response to these bacteria, but the specific genes within these families differed, including lysozyme, aspartyl protease, saposin-like, and C-type lectin genes (*Figure 6D–F*, *Figure 6—figure supplement 1A, B, D, and E*). These results suggest that DBL-1 is involved in regulating (positively and negatively) transcription of not only some innate immunity genes specific to the bacterial challenge, but also genes that are commonly regulated upon exposure to a variety of bacteria. This supports a role for DBL-1 signaling in differentially regulating molecular host responses to the bacterial challenge.

## DBL-1 differentially regulates expression of innate immunity genes specific to the Gram nature of bacteria

To determine if the expression of candidate target immunity genes is regulated by DBL-1 signaling in response to a wider variety of bacterial exposures, we used reporters of select immunity-related genes. Based on our RNA-sequencing dataset and previously published reports, we shortlisted genes based on their differential expression in either Gram-negative or Gram-positive-challenged animals

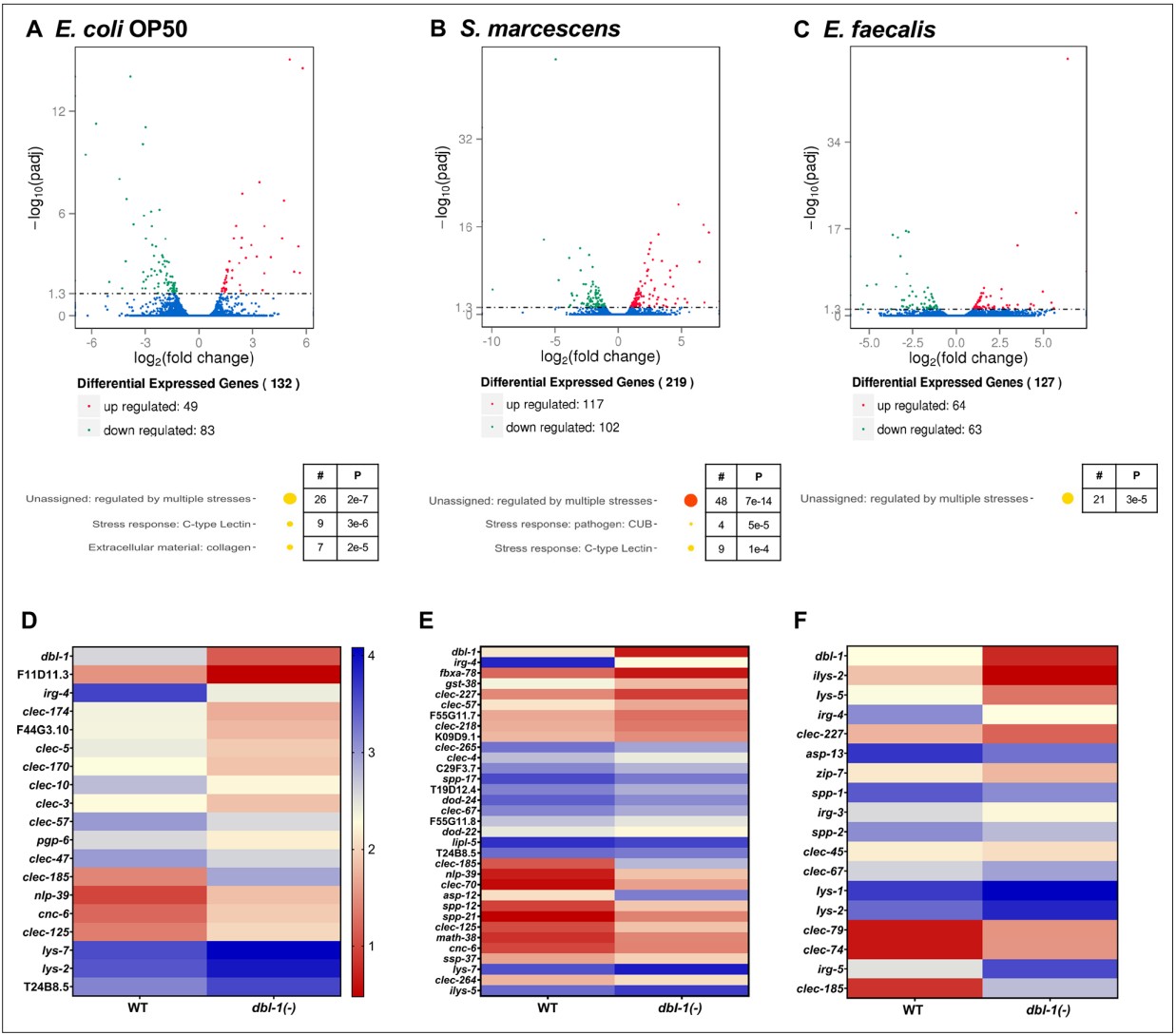

**Figure 6.** DBL-1 regulates differential gene expression in response to Gram-negative and Gram-positive bacteria. Wild-type and *dbl-1(-)* animals were exposed to *E. coli* OP50 (control), *S. marcescens*, or *E. faecalis* for 2 d starting at the L4 stage. RNA-seq analysis using volcano plots shows differential gene expression in animals lacking DBL-1 exposed to (**A**) *E. coli* OP50, (**B**) *S. marcescens*, and (**C**) *E. faecalis* in comparison to wild-type animals exposed to the same bacteria (adjusted p-value<0.01). Genes downregulated in *dbl-1(-)* animals are represented in green, genes upregulated in *dbl-1(-)* animals are represented in red, and genes with no change in expression are represented in blue in the volcano plots. Functional categories of differentially expressed genes (WormCat Category 3) are shown. The circle size and color represent relative number of genes in each group and p-value, respectively (orange, p<10⁻¹⁰; yellow, p<0.05; see *Figure 6—source data 1* for details). Heatmaps show differential innate immunity gene expression in animals lacking DBL-1 exposed to (**D**) *E. coli* OP50, (**E**) *S. marcescens*, and (**F**) *E. faecalis* in comparison to wild-type animals exposed to the same bacteria. Average log FPKM values from three independent trials are represented.

The online version of this article includes the following source data and figure supplement(s) for figure 6:

**Source data 1.** Related to *Figure 6*.

**Figure supplement 1.** DBL-1-dependent and DBL-1-independent differential gene expression in response to Gram-negative and Gram-positive bacteria.

**Figure supplement 1—source data 1.** Related to *Figure 6—figure supplement 1*.

(or both) and their requirement for DBL-1 signaling (or not) for their differential expression. This list of genes included *dod-22*, F55G11.7, *irg-4*, *dod-24*, and *ilys-3*. We measured and compared intestinal expression of these reporter genes in wild-type and *dbl-1(-)* backgrounds exposed to control or test bacteria. Animals were grown on *E. coli*, synchronized as L4s, and fed on control *E. coli*, Gram-negative, or Gram-positive bacteria for 48 hr before expression of these selected genes was measured and compared. *dod-22* is a gene that is known to be regulated by the insulin-like signaling

pathway transcription factor DAF-16 (**Murphy et al., 2003**). It is expressed primarily in the intestine and is involved in defense response to Gram-negative bacteria (**Alper et al., 2007**). RNA-sequencing analyses revealed *dod-22* expression was induced in the wild type, but not in *dbl-1(-)* animals, only upon exposure to *S. marcescens* (**Figure 6—source data 1**). The *dod-22* reporter was induced in the wild-type background in the presence of all Gram-negative test bacteria compared to the control but was not induced in response to the panel of Gram-positive bacteria. This induction of the *dod-22* reporter on Gram-negative test bacteria was lost in the *dbl-1(-)* background, though loss of DBL-1 did not affect expression levels on the control *E. coli. dod-22* reporter fluorescence remained largely unchanged in *dbl-1(-)* animals exposed to Gram-positive test bacteria (**Figure 7A and B**). These results indicate that DBL-1 is not required for the basal expression of *dod-22* but is required for *dod-22* induction on Gram-negative bacteria, which correlated with the RNA-seq data.

F55G11.7 is majorly expressed in the intestine and has been reported to be involved in innate immune responses to both Gram-negative and Gram-positive bacteria in *C. elegans*. Previous findings indicate that F55G11.7 is regulated by insulin, MAPK, and DBL-1 signaling pathways (**Alper et al., 2007**). Expression of F55G11.7 was reduced in *dbl-1(-)* mutants compared to wild-type fed on *S. marcescens* in our RNA-seq data (**Figure 6—source data 1**). Expression of the F55G11.7 reporter did not consistently change upon exposure to all test bacteria in the wild-type background except *B. megaterium*, where reporter expression was reduced (in two of three trials). We observed a significant reduction of F55G11.7 reporter activity in animals lacking DBL-1 except in response to *B. megaterium* and *E. faecalis* compared to wild-type animals. Additionally, a further reduction of F55G11.7 reporter fluorescence in *dbl-1(-)* animals was observed in response to Gram-negative *S. marcescens* and Gram-positive *B. megaterium* and *S. epidermidis* compared to the response on control bacteria (**Figure 7C and D**). These findings indicate that expression of F55G11.7 is not induced upon exposure to most of the test bacteria but is generally downregulated upon loss of DBL-1, expanding on previous findings that F55G11.7, which is induced upon *S. marcescens* Db11 and *P. aeruginosa*, is regulated by DBL-1 in response to pathogenic bacteria (**Alper et al., 2007**).

*irg-4* is expressed in intestine and head neurons (**Alper et al., 2007**). It is known to be involved in defense responses to Gram-negative bacteria and has been shown to be regulated by insulin, MAPK, and DBL-1 signaling pathways (**Shapira et al., 2006**; **Troemel et al., 2006**; **Nandakumar and Tan, 2008**; **Peterson et al., 2019**). Based on our RNA-sequencing data, expression of *irg-4* in wild-type animals was induced upon exposure to *S. marcescens* but was reduced in response to *E. faecalis* in comparison with the control bacteria. Furthermore, we found *irg-4* to be a DBL-1-responsive gene. Expression of *irg-4* was reduced in *dbl-1(-)* compared to wild-type animals fed on control, *S. marcescens*, and *E. faecalis* (**Figure 6—source data 1**). In the wild-type background, we observed a visible, reproducible induction of *irg-4* reporter activity in response to Gram-negative *S. marcescens*, but not to *K. oxytoca* or *E. cloacae* (**Figure 7E**). Expression of the *irg-4* reporter in wild-type animals is reproducibly unaltered in response to Gram-positive bacteria (**Figure 7F**). Loss of DBL-1 did not reproducibly alter *irg-4* reporter activity in control conditions. However, *irg-4* reporter induction in response to Gram-negative *S. marcescens* was lost in the *dbl-1(-)* background (**Figure 7E and F**). Indeed, *irg-4* reporter expression was reduced compared to the wild-type background on all test bacteria except Gram-negative *E. cloacae* and Gram-positive *E. faecalis*. These results support that *irg-4* is responsive to a broad range of bacteria and is regulated in part by DBL-1 signaling.

*dod-24* is mainly expressed in the intestine (**Yamawaki et al., 2010**). *dod-24* is regulated by the insulin-like signaling transcription factor DAF-16 in response to Gram-negative bacteria (**Shapira et al., 2006**). RNA-sequencing data indicated that *dod-24* expression in wild-type animals was induced in response to *S. marcescens* and dampened in response to *E. faecalis* compared to wild-type animals fed on control bacteria. These changes in *dod-24* expression were lost in *dbl-1(-)* animals (**Figure 6—source data 1**). We observed robust expression of *dod-24* reporter activity in all tested Gram-negative bacteria, including the control, *E. cloacae*, *K. oxytoca*, and *S. marcescens*. Furthermore, *dod-24* reporter activity was greater in response to *E. cloacae* and *S. marcescens* than to the control bacteria (**Figure 7G**). We observed a striking decrease of *dod-24* reporter activity in wild-type animals exposed to all tested Gram-positive bacteria (*B. megaterium*, *E. faecalis*, and *S. epidermidis*) compared to the control bacteria (**Figure 7H**). Loss of DBL-1 resulted in a significant reduction of *dod-24* reporter activity in control conditions. *dod-24* reporter activity was also drastically reduced in all tested Gram-negative bacterial conditions to levels significantly lower than the wild type (**Figure 7G**).

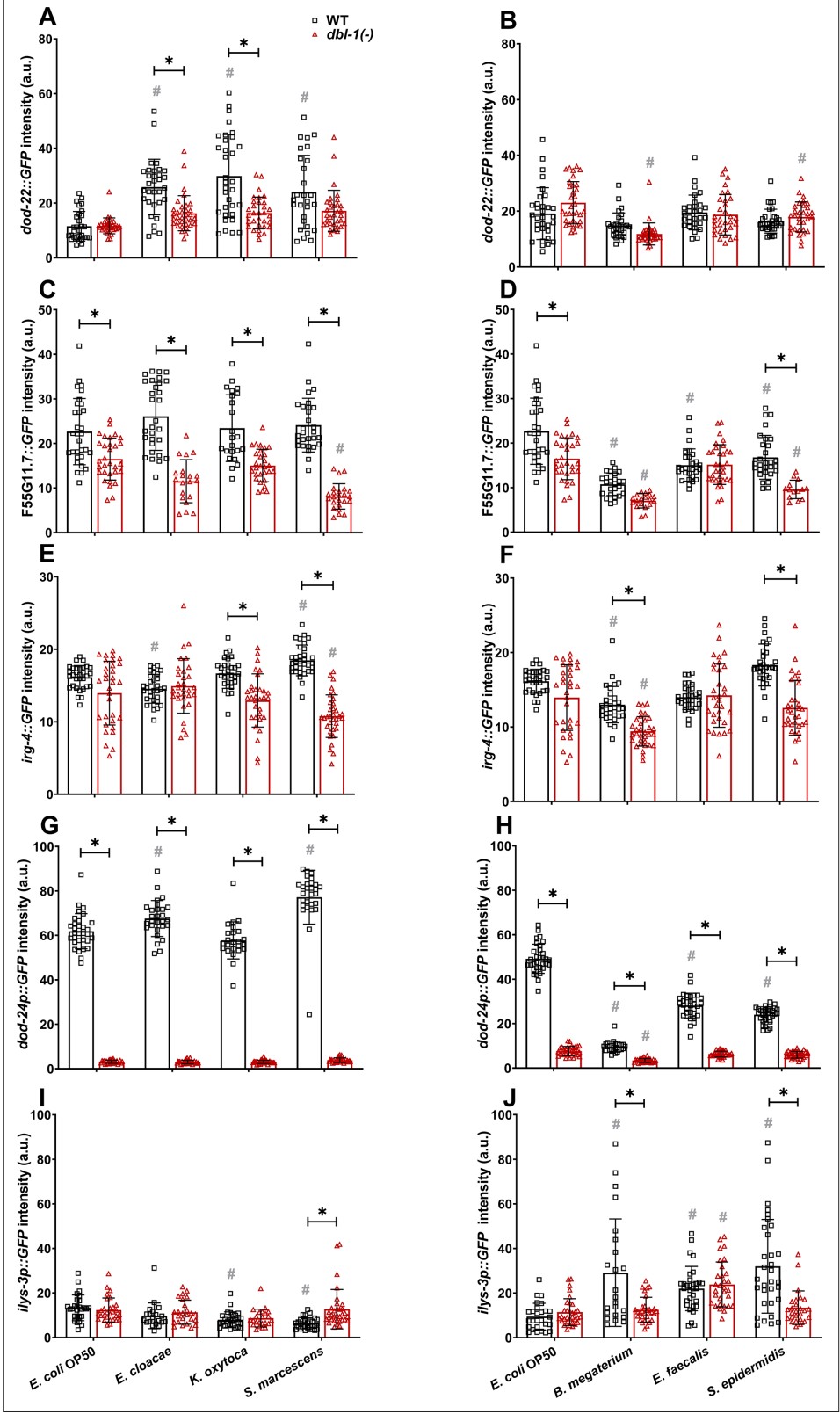

**Figure 7.** Innate immune reporter activity is regulated by exposure to specific bacteria and by DBL-1 signaling. Comparison of (**A, B**) *dod-22*::GFP, (**C, D**) F55G11.7::GFP, (**E, F**) *irg-4*::GFP, (**G, H**) *dod-24p*::GFP, and (**I, J**) *ilys-3p*::GFP intensities in adult wild-type and *dbl-1(-)* animals after a 2-day exposure to the following bacteria; control *E. coli* OP50, *E. cloacae*, *K. oxytoca*, *S. marcescens*, *B. megaterium*, *E. faecalis*, or *S. epidermidis*. Three

*Figure 7 continued on next page*

*Figure 7 continued*

independent trials were performed. One representative trial is shown. Error bars represent standard deviation. n ≥ 14 per condition in each trial. **p<0.05 compared to wild-type animals exposed to the same bacteria, and #p<0.05 respective genotype exposed to test bacteria in comparison to control bacteria, by two-way ANOVA using Tukey's post hoc test.

The online version of this article includes the following source data for figure 7:

**Source data 1.** Related to *Figure 7*.

In the three Gram-positive conditions, loss of DBL-1 resulted in a further decrease of *dod-24* reporter fluorescence relative to the wild type (*Figure 7H*). These results confirm that *dod-24* is differentially expressed in response to an array of Gram-negative and Gram-positive bacteria and indicate that DBL-1 signaling plays a major role in regulating *dod-24* expression.

*ilys-3*, which encodes a lysozyme, is expressed in the intestine, pharynx, and coelomocytes. It is induced in response to Gram-positive bacteria (*Gravato-Nobre et al., 2005*). By RNA-sequencing, we confirmed expression of *ilys-3* was induced in wild-type animals exposed to Gram-positive *E. faecalis* but not Gram-negative *S. marcescens*. This induction was lost in *dbl-1(-)* mutants (*Figure 6—source data 1*). *ilys-3* reporter activity in wild-type animals was unchanged or reduced in response to the tested Gram-negative bacteria compared to the control (*Figure 7I*). We observed induction of *ilys-3* reporter activity upon exposure to all tested Gram-positive bacteria (*B. megaterium*, *E. faecalis*, and *S. epidermidis*) (*Figure 7J*). Loss of DBL-1 did not alter *ilys-3* reporter activity in control conditions. The *ilys-3* reporter activity remained at relatively low levels in animals lacking DBL-1 exposed to test Gram-negative bacteria (*Figure 7I*). However, *ilys-3* reporter activity also remained at relatively low levels upon loss of DBL-1 in response to Gram-positive bacteria *B. megaterium* and *S. epidermidis* but was wild type in response to *E. faecalis* (*Figure 7J*), consistent with our RNA-seq results. These results suggest that while DBL-1 is not required for basal levels of *ilys-3* expression, it is required for the induction of *ilys-3* expression in response to some Gram-positive bacteria.

Together, these results expand on the RNA-sequencing dataset and validate some of the RNA-sequencing results. These data provide a snapshot of the gene expression profile of the host upon exposure to a variety of bacteria. These findings collectively indicate that DBL-1 signaling also tailors defense responses to specific bacteria by regulating antimicrobial gene expression.

## Discussion

Animals are subjected to a range of bacterial challenges, and how they perceive these potential threats and respond to them can be critical to the animals' health. Understanding how hosts respond to different bacteria is important for developing therapeutic strategies to help fight infections and prevent diseases. Our work expands the current understanding of how the model organism *C. elegans* integrates an arsenal of responses—from the molecular to the organismal—to different bacteria, and identifies roles for the DBL-1/TGF-β pathway in robust host-specific responses to different types of bacteria. For this work, we established a panel of human opportunistic pathogens, including three Gram-negative and three Gram-positive strains, for the study of long-term protective responses in the roundworm *C. elegans*. The bacteria and conditions we used elicit unique host response patterns that allowed us to interrogate the role of the DBL-1 pathway in responding to different bacterial exposures. Our work indicates that the choice of test bacteria is crucial in determining involvement of signaling pathways in host defenses because different host responses are elicited by different bacteria. While DBL-1 has a known role in transcriptionally regulating innate immune gene expression, we show that the specific defense responses mediated by DBL-1 are not only molecular but are also behavioral.

Our results support a model that the DBL-1 signaling pathway influences the organisms' perception of bacterial threat and helps keep the host defenses in check. Animals with reduced DBL-1 signaling—whether by downregulating signaling or by mutation—may perceive many environments (even standard lab conditions) as more threatening. Animals lacking DBL-1 also display a reduced lifespan in response to the Gram-negative bacteria that are sensed to be more harmful (*Figure 1*). These animals also mildly reduce intake of select Gram-negative bacteria, another way to reduce animals' interaction with the potential threat (*Figure 2*). DBL-1 pathway mutants display an outsized avoidance response to the Gram-negative bacteria in the test panel (*Figure 3*). There is also a correlation between the

reduced feeding and extension of lifespan observed in animals exposed to select bacteria. Notably, loss of DBL-1 did not result in an avoidance response to Gram-positive bacteria in the panel. However, we suspect that because exposure to the Gram-positive bacteria reduced DBL-1 signaling activity, the avoidance response did not alter dramatically in animals lacking DBL-1. The intake of the Gram-positive bacteria was reduced in wild-type populations which was further reduced in *dbl-1(-)* populations. While the reduced feeding observed in wild-type animals fed Gram-positive bacteria correlated with their extended lifespan, it did not correlate completely with the lifespan of *dbl-1(-)* populations. Lifespan of *dbl-1(-)* populations on the tested Gram-positive bacteria was shorter than the wild-type populations. Neither damage to the intestine nor bacterial colonization was the underlying cause for the DBL-1-mediated lifespan alterations in response to the tested bacteria.

Determining how DBL-1 is involved in such behavioral modifications, especially avoidance behavior, in response to different bacteria warrants future investigation. It will be of interest to determine the bacterial cues that are responsible for eliciting DBL-1-mediated avoidance behavior. A previous study has reported that animals respond to and avoid a biosurfactant, serrawettin W2, produced by *S. marcescens*, using the AWB chemosensory neuron (*Pradel et al., 2007*). It will also be of interest to determine the neuronal pathways involved in processing these sensory cues into organismal responses. DBL-1 is required for inhibiting avoidance behavior and for aversive learning, but the cells involved in both DBL-1 secretion and reception are different for these two behaviors (*Zhang and Zhang, 2012*; *Olofsson, 2014*). ASH and ASI chemosensory neurons control innate immune responses and ASI promotes pathogen avoidance behavior (*Cao et al., 2017*). DBL-1 secreted from the AVA interneurons activates DBL-1 signaling in the hypodermis to regulate aversive learning upon exposure to Gram-negative *P. aeruginosa* (*Zhang and Zhang, 2012*). The AWB, ASH, and ASI sensory neurons are connected to the AVA interneuron through other sensory neurons and interneurons, suggesting possible neuronal circuits that mediate DBL-1-dependent avoidance responses (*Chen et al., 2006*).

In addition, could the DBL-1 pathway be interacting with other molecular regulators to respond to bacterial cues? A genetic interaction was identified between the DBL-1 pathway and the DAF-7/TGF-β-like pathway in the context of dauering, a developmental response to reproductively adverse environmental conditions (*Krishna et al., 1999*; *Morita et al., 1999*; *Maduzia et al., 2005*). DAF-7 also contributes to the perception of and avoidance of *P. aeruginosa* (*Meisel et al., 2014*; *Singh and Aballay, 2019*). Future work will determine if DBL-1 affects avoidance behavior by repressing DAF-7 signaling. Another possibility is that DBL-1 signaling responds to physiological changes caused by bacteria; distension of the intestinal lumen by bacterial colonization has been shown to elicit avoidance behaviors (*Singh and Aballay, 2019*; *Filipowicz et al., 2021*).

Another finding of this work is the requirement of the Smad machinery to mediate specific avoidance responses to our panel of bacteria (*Figure 3*). Olofsson previously showed that loss of DBL-1, SMA-2, or SMA-4 increases avoidance of *E. coli* (*Olofsson, 2014*). While some of our results with *sma-4* are similar to theirs, we did not observe a robust avoidance of *E. coli* upon loss of DBL-1 or SMA-2, which may be caused by differences in growth conditions. However, our work demonstrates that canonical DBL-1 signaling strongly suppresses avoidance to Gram-negative bacteria, but not to Gram-positive bacteria (*Figure 3*). Furthermore, mild DBL-1-independent induction of *sma-4* expression in response to Gram-positive bacterial conditions was also observed (*Figure 4*). Lastly, the SMAD reporter expression is not completely lost in animals lacking DBL-1 upon exposure to test Gram-negative bacteria, which also provides evidence for Smad activation by a signaling pathway other than DBL-1. These results suggest that SMA-4 maybe recruited for molecular defenses by something other than DBL-1. Interestingly, ATF-7, a transcription factor activated by PMK-1/MAPK, is required for downregulation of *sma-4*—but not other DBL-1 pathway component genes—in wild-type animals exposed to Gram-negative *Pseudomonas aeruginosa* PA14 (*Fletcher et al., 2019*). In addition, SMA-4 is predicted to genetically interact with PMK-1/MAPK (*Zhong and Sternberg, 2006*). It will be of interest to determine if MAPK signaling or another innate immune pathway affects Smad signaling.

DBL-1 signaling is also important for coordinating molecular responses to both Gram-negative and Gram-positive bacteria. Additionally, we also discovered that DBL-1 signaling itself is altered in response to Gram-negative and Gram-positive bacteria. Regulation of DBL-1 signaling is part of the host's molecular response: the Gram-negative bacteria of our panel induced DBL-1 signaling while the Gram-positive bacteria repressed DBL-1 signaling. However, within these two bacterial groups, the host responses were tailored to the specific bacterial challenge (*Figure 5*). Our results implicate

DBL-1 signaling as an important part of the molecular antimicrobial 'fingerprint' proposed by the Schwartz lab (*Alper et al., 2007*). Our RNA-sequencing analyses indicate both common and unique transcriptome-wide alterations mediated by DBL-1 in adult animals after 2 d of exposure to Gram-negative and Gram-positive bacteria (*Figure 6*). We find that DBL-1 signaling is involved in activating as well as repressing innate immunity genes to maintain a balance of host immune responses (to possibly avoid overactivation of host immune responses). Some genes identified in the RNA-seq analysis and used in the reporter studies have previously been shown to be targets of other innate immune response pathways, suggesting that these genes are regulated by multiple molecular signaling pathways, possibly in a context-dependent manner. Our reporter studies highlight DBL-1-responsive and DBL-1-independent target genes that are differentially regulated depending on the bacterial source (*Figure 7*). Furthermore, DBL-1 signaling activity was induced in response to Gram-negative bacteria, and it further regulated expression of unique downstream known and putative antimicrobial genes. In contrast, even though the DBL-1 signaling activity was strikingly repressed in response to the tested Gram-positive bacteria, DBL-1 was still required to regulate expression of target immunity genes. While many gene classes differentially regulated by specific pathogens have been previously identified as important innate immune response genes, our work highlights the role that DBL-1 plays in tailoring the molecular responses *C. elegans* engages against a range of bacteria (*Wong et al., 2007*; *Alper et al., 2007*; *Gravato-Nobre et al., 2005*; *Shapira et al., 2006*; *Engelmann et al., 2011*; *Troemel et al., 2006*).

Overall, we propose that loss of DBL-1 signaling changes the animal's perception of the environment as more hostile, and this results in more robust protective responses that depend on the specific bacterial challenge. In this model, neuronally secreted DBL-1 targets hypoderm, intestine, and pharynx to regulate host defense responses. SMA-4 plays a double role in innate immune responses, acting as part of the core DBL-1 signaling pathway but also acting in another additive way, suggesting crosstalk with other signaling pathways. Future work may identify crosstalk between the DBL-1 pathway and other molecular pathways in regulating organismal defense responses. In summary, these findings support a central role for DBL-1/TGF-β signaling not only in crafting tailored responses to unique bacterial challenges, from transcription of specific innate immunity genes to behavioral responses, but also being targeted in response to bacteria.

# Materials and methods
## Strains and maintenance
### *C. elegans* strains
All *C. elegans* strains were maintained on EZ media plates at 20°C (0.55 g Tris-Cl, 0.24 g Tris base, 3.1 g BD Bacto Peptone, 8 mg cholesterol, 2 g sodium chloride, 20 g agar, in water to 1 ll) (*Madhu et al., 2019*). *C. elegans* strains were maintained without contamination or starvation for at least five generations before every experiment. *Supplementary file 3* includes the list of all strains used in this study. These strains were generated by standard genetic crosses and confirmed by small body size phenotype and presence of fluorescence.

### Bacterial strains
The bacterial strains used in this study include *B. megaterium* (ATCC 14581, Carolina Biological Supply Company), *E. coli* (OP50), *E. cloacae* (ATCC 49141), *E. faecalis* (ATCC 51299), *K. oxytoca* (ATCC 49131), *S. marcescens* (D1, Carolina Biological Supply Company), and *S. epidermidis* (ATCC 49134). *E. faecalis* in brain heart infusion media and all other bacteria in tryptic soy broth were grown for 9 hr to stationary phase at 37°C as previously described (*Madhu et al., 2019*). Bacterial cells were pelleted at 5000 rpm for 15 min and concentrated 20-fold. EZ media plates, which support *C. elegans* growth, were freshly seeded with concentrated bacteria in full lawns. The seeded plates were incubated at 37°C overnight before use in experiments.

## Lifespan assay
Lifespan assay was performed as previously described (*Sifri et al., 2003*; *Reddy et al., 2009*; *Amrit et al., 2014*). Concentrated bacterial cultures were spread on 6-cm-diameter EZ media plates (full lawn plates) containing 50 µg/ml FUdR to cover the surface of the plates entirely. FUdR, which prevents

reproduction by blocking DNA synthesis, was used to prevent offspring from confounding the scoring. Wild-type and *dbl-1(-)* animals (n ≥ 30) were fed on control and test bacteria on full lawn plates at the L4 stage in quadruplicate. The plates were scored for live and dead nematodes every 24 hr until all animals were dead. Animals were scored as dead if they did not respond to gentle touch with a sterilized platinum wire and were removed from the plate. At least three independent trials were performed. Worms that died by desiccating on the walls of the plates were censored from the analysis.

## Intestinal barrier function assay

The intestinal barrier function assay was performed as previously described (*Kissoyan et al., 2019*). Wild-type and *dbl-1(-)* L4 animals were fed on control and test bacteria on full lawn plates. The assay was performed when about 50% of the population with the lowest mean lifespan remained alive. At least 15 animals were sampled at the specified times to examine intestinal tissue integrity. The intestinal barrier integrity was assessed using a blue dye, erioglaucine disodium salt (5% wt/v), as an indicator of tissue integrity as the animals age. Leaking of this blue dye outside the intestinal lumen indicates reduced intestinal integrity. The animals were washed with S buffer and were incubated in erioglaucine disodium salt solution in a 1:1 ratio for 3 hr. The animals were then washed thrice with S buffer and were mounted on 2% agarose pads on glass slides. 10 µM levamisole was added to paralyze the animals. The animals were imaged on a Nikon DS-Ri2 camera mounted on a Nikon SMZ18 dissecting microscope. The leakiness of the intestine was assessed and scored for no leakage/no Smurf, mild leakage/mild Smurf, and severe leakage/severe Smurf phenotypes. The experiment was performed in at least three independent trials for each experimental condition.

## Bacterial colonization assay

The bacterial colonization assay was performed using a method adapted from *Portal-Celhay et al., 2012*; *Eng and Nathan, 2015*. Wild-type and *dbl-1(-)* L4 animals were fed on *S. marcescens* on full lawn plates. The assay was performed when about 50% of the population with the lowest mean lifespan remained alive. Five animals in quadruplicates were picked and washed in 5 µl drops of M9 buffer containing 25 mM levamisole to paralyze animals and inhibit their pharyngeal pumping and expulsion. The animals were washed with thrice more with M9 buffer. The washed animals were placed in 1.5 ml Eppendorf tubes containing 120 µl M9 buffer and 100 µl was aliquoted from the tube to and incubated at 37°C overnight on tryptic soy agar plates to yield background bacterial counts. To the animals remaining in the tube, 30 µl M9 buffer containing 1% Triton X-100 was added and the animals were mechanically disrupted by using a pestle. This homogenized worm lysate was incubated overnight at 37°C on tryptic soy agar plates to yield bacterial counts. Bacterial colony-forming units (CFUs) per worm were calculated using the formula: (colony number × dilution factor)/ (5 animals × 100 µl lysate plated). Finally, CFU per worm of the background was subtracted from the CFU per worm of the worm lysate to compare bacterial colonization between wild-type and *dbl-1(-)* populations.

## Pharyngeal pumping rate

Wild-type and *dbl-1(-)* L4 animals (n = 12) were fed on control and test bacteria on full lawn plates. The number of contractions of the pharyngeal bulb was counted for 20 s to calculate the pharyngeal pumping rate of animals. Two counts were made and averaged for each animal. Three independent trials were performed in triplicate (*Clark et al., 2018*).

## Microbial avoidance assay

Microbial avoidance assays were performed as previously described (*Chang et al., 2011*). Then, 20 µl of the concentrated bacterial cultures were spotted on 6 cm diameter EZ media plates and incubated at 37°C overnight. Wild-type, *dbl-1(-)*, *sma-2(-)*, *sma-3(-)*, and *sma-4(-)* L4 hermaphrodites were placed on control and test bacterial lawn (n = 30 per condition/trial, performed in triplicate). The plates were scored for number of worms occupying the lawn at the indicated time points. Three independent trials were performed. The avoidance ratio (A) was calculated using the formula: A = number of animals off the lawn/total number of animals.

## RNA isolation

Animals were synchronized as embryos by bleaching mixed-stage populations (*Stiernagle, 2006*). Total RNA was extracted from animals at 48 hr after the L4 stage. Total RNA was extracted by the freeze cracking method as previously described (*Portman, 2006*).

## Differential expression analysis by RNA sequencing

RNA from wild-type and *dbl-1(-)* adult populations fed on control and test bacteria was extracted in three independent trials. Sequencing libraries from the extracted RNA were generated using the NEBNext RNA Library Prep Kit for Illumina (NEB, USA) following the manufacturer's recommendations. Also, 1 µg RNA of each sample was used as input material for the RNA sample preparations. Novogene performed RNA sequencing of samples. Differential expression analysis of wild-type compared to *dbl-1(-)* populations grown on different bacteria was performed using the DESeq R package (1.18.0) (*Love et al., 2014*). Genes with an adjusted p-value<0.01 found by DESeq were assigned as differentially expressed.

## cDNA synthesis and qRT-PCR

After RNA isolation, cDNA was synthesized and quantitative real-time PCR was performed as previously described (*Madhu et al., 2020*). 2 µg of total RNA isolated was primed with oligo(dT) and reverse transcribed to yield cDNA using the SuperScript III reverse transcriptase kit as per the manufacturer's protocol (Invitrogen). Real-time PCR was performed on a QuantStudio3 system (Applied Biosystems by Thermo Fisher Scientific) using the PowerUP SYBR Green master mix (Applied Biosystems) according to the manufacturer's instructions. Three independent biological trials were performed. Each biological trial was performed in three technical replicates for each condition. Primer sequences are available in *Supplementary file 4*. QuantStudio Design and Analysis Software v1.5.1 was used to calculate raw $C_t$ values. The $C_t$ values for the target genes were normalized to the housekeeping gene *act-1* (actin) (Applied Biosystems by Thermo Fisher Scientific). Fold change in gene expression between experimental sample and the control was determined by using the $2^{(-\Delta\Delta C_t)}$ method.

## Imaging

RAD-SMAD reporter strains were placed on full lawns of the control or test bacteria at the L4 stage. L2 progeny were mounted on 2% agarose pads and anesthetized by using 1 mM levamisole, and fluorescence was captured by a Zeiss LSM 900 confocal microscope using a 40× oil objective. At least 10 L2 animals with at least 5 hypodermal nuclei per worm in the focal plane were imaged per condition when fluorescence was visible, giving a moderate effect size as determined by power analysis. The experiment was performed in three independent trials. The microscope conditions were optimized with respect to the control and test conditions and kept consistent within each trial. Mean fluorescence intensities of the five hypodermal nuclei in each animal were calculated, and the average intensity of each strain was determined as previously described using Zeiss ZEN lite software (*Savage-Dunn et al., 2019*). Data was statistically analyzed by unpaired *t*-test.

The innate immune reporter strains were transferred to full lawns of the control and test bacteria at the L4 stage and were imaged after 48 hr of exposure. Fluorescence of the reporter strains was captured by a Nikon DS-Ri2 camera mounted on a Nikon SMZ18 dissecting microscope. Animals were mounted on 2% agarose pads and anesthetized with 1 mM levamisole. At least 15 animals were imaged per condition as determined by power analysis with a moderate effect size. The microscope conditions were optimized with respect to the control and test conditions and kept consistent within each trial. However, imaging exposure times were different between some trials to prevent saturation of signal in experimental conditions. Three independent trials were performed. Mean intestinal fluorescence intensities were measured using the Nikon NIS Elements AR v5.02 software.

## Statistical analyses

Lifespans of *C. elegans* populations were calculated by the Kaplan–Meier method, and statistical analysis was performed using the two-tailed log-rank test. The average pharyngeal pumping rates were analyzed using two-way ANOVA using Tukey's post hoc test. Bacterial colonization counts (CFU per worm) were compared using the two-tailed unpaired *t*-test. The intestinal barrier function phenotypes and percentages of animals expressing RAD-SMAD reporter exposed to

Gram-positive bacteria were statistically analyzed by the chi-square test. The avoidance ratio was compared by repeated measures ANOVA using Tukey's post hoc test. qRT-PCR values and mean fluorescence intensities were evaluated using ANOVA with Dunnett's multiple-comparisons test. RNA-sequencing analysis was performed using the DESeq R package (1.18.0). The resulting p-values were adjusted using the Benjamini and Hochberg's approach for controlling the false discovery rate.

## Acknowledgements

We thank Rosylin Roy and Neethu G Issac for technical assistance. We thank James Lundgren and TWU's Center for Research Design and Analysis for assistance with statistical analyses. Some bacterial strains were provided by Amy Jo Hammett. Some strains were obtained from the Caenorhabditis Genetics Center (CGC), which is funded by the NIH Office of Research Infrastructure Programs (P40 OD010440). We thank WormBase. We thank Laura Hanson and Gumienny lab members for constructive feedback. Novogene provided *Figure 6A–C* and *Figure 6—figure supplement 1A, B, D, and E*. This work was supported by NIH R01GM097591 to TLG, Jane Nelson Institute for Women's Leadership funding to TLG, TWU Research Enhancement Program funding to TLG, internal funding by Texas Woman's University to TLG, and TWU Experiential Learning Scholar Awards to BM. This work is dedicated to Rita and Jay Madhu.

## Additional information

### Funding

| Funder | Grant reference number | Author |
|---|---|---|
| Office of Extramural Research, National Institutes of Health | R01GM097591 | Tina L Gumienny |
| Jane Nelson Institute for Women's Leadership | internal grant | Tina L Gumienny |
| Texas Woman's University | TWU Research Enhancement Program (faculty grant) | Tina L Gumienny |
| Texas Woman's University | TWU Experiential Learning Scholar Award (student funding) | Bhoomi Madhu |

The funders had no role in study design, data collection and interpretation, or the decision to submit the work for publication.

### Author contributions

Bhoomi Madhu, Conceptualization, Data curation, Formal analysis, Supervision, Funding acquisition, Validation, Investigation, Visualization, Methodology, Writing – original draft, Writing – review and editing; Mohammed Farhan Lakdawala, Data curation, Validation, Investigation, Writing – review and editing; Tina L Gumienny, Conceptualization, Resources, Data curation, Formal analysis, Supervision, Funding acquisition, Writing – original draft, Project administration, Writing – review and editing, Visualization

### Author ORCIDs

Bhoomi Madhu http://orcid.org/0000-0002-8931-5345
Mohammed Farhan Lakdawala http://orcid.org/0000-0003-2271-2255
Tina L Gumienny http://orcid.org/0000-0002-3932-7815

### Decision letter and Author response

Decision letter https://doi.org/10.7554/eLife.75831.sa1
Author response https://doi.org/10.7554/eLife.75831.sa2

## Additional files

### Supplementary files
- Supplementary file 1. Summary of survival assay reported in *Figure 1*.
- Supplementary file 2. Fraction of animals expressing detectable RAD-SMAD reporter fluorescence in *dbl-1(-)* animals upon exposure to control and test Gram-negative and Gram-positive bacteria.
- Supplementary file 3. List of strains used and created for this work.
- Supplementary file 4. List of primers used for qRT-PCR.
- Transparent reporting form

### Data availability
All data generated or analysed during this study are included in the manuscript and supporting file.

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
