## [Editor Report]

This study provides valuable insight into the role of the TGF-β signaling pathway in the immune response of *C. elegans*. The authors report a convincing analysis of molecular and behavioral responses to a broad panel of bacteria, dissecting the contribution of the TGF-β pathway to these responses.

---

## [Decision Letter]

[Editors' note: this paper was reviewed by Review Commons.]

Thank you for submitting your article "The DBL-1/TGF-β signaling pathway tailors behavioral and molecular host responses to a variety of bacteria in *Caenorhabditis elegans*" for consideration by *eLife*. Your article has been reviewed by 3 peer reviewers at Review Commons, and the evaluation at *eLife* has been overseen by a Reviewing Editor and Piali Sengupta as the Senior Editor.

Based on the reviewers' comments, we are potentially interested in this work. However, the editors agree that the proposed revision plan is inadequate as presented. Moreover, the editors raised a few additional issues that will also need to be addressed with new data and/or experiments for reconsideration at *eLife*.

Specifically:

1. (Issue also raised by Reviewer 1) Figure 3: While it is appreciated that the RT-PCR experimental design quantified by gene and not by bacteria, it is nevertheless important that a subset of these data be repeated by bacterial treatment.

2. Related to the above in terms of data organization and presentation: An important conclusion of the work is that SMA-4 acts independently of the canonical DBL-1 signaling pathway. Avoidance responses are compared in dbl-1(0) and sma mutants across time in Figures 2 and 3. However, given the variability of these assays it is difficult to compare across independent experiments. sma-4 and dbl-1 mutants should be tested in parallel in the same experiments and the data plotted in the same graph.

3. (Issue also raised by Reviewer 3) – given the importance of the sma-4 data, data using another independent sma-4 allele need to be presented in the main figures.

4. (Issues raised by all reviewers) – In order for the reviewers and editors to review the data, all RNASeq data need to be included in the manuscript for review. The data should also be appropriately analyzed and should include standard and complete information including gene names, read counts, and other statistics. In the absence of the actual data, the RNASeq analyses (including the sunburst plot) cannot be appropriately reviewed, and the justification for focusing on specific genes cannot be assessed.

5. (Issue also raised by Reviewer 1) – In several figures (including Figure 5 and Figure 6) – the statistics should include a correction for multiple comparisons.

---

## [Author Response]

1. General statements:

Thank you for the three reviews. The reviewers’ comments showed that they grasped the goals of this work and its context in the field. Their comments were fair and helped us further improve our manuscript.

2. Point-by-point description of the revisions:

Reviewer #1Major comment:Overall, the key conclusions are convincing and reflect the presented results. However, one conclusion I find too strong given the lack of presented data:According to the RAD-SMAD reporter DBL-1 signaling is presumably repressed when worms are exposed to Gram-positive bacteria. This observation is solely based on the data set with percentages presented in Table S4. The fact that there are no images (even with weak signals!) nor appropriate statistics presented as well as the high variability among the trials (e.g. WT on *E. faecalis* ranges from 21 to 95%) does not convince enough for such a strong statement.

We do present images of animals expressing the RAD-SMAD reporter on the panel of bacteria in Figure 6. In populations of animals expressing this reporter, 100% consistently express it in the control condition, so the range of animals fed on *E. faecalis* that still express this reporter, while variable (5–79% of the population), shows that this reporter is consistently repressed in the population on *E. faecalis*. However, because of the variability of animal responses on *E. faecalis*, we rephrased our statement to indicate that strong pathway repression is observed in response to some Gram-positive bacteria instead of pathway repression is seen across all Gram-positive bacteria tested.

Please reconsider the emphasis on the conclusion for Gram-positive bacteria, e.g. "Animals activate DBL-1 pathway activity in response to Gram-negative bacteria and repress it in response to Gram-positive bacteria, demonstrating bacteria-responsive regulation of DBL-1 signaling." (e.g. from the abstract).

We rephrased this abstract sentence to:

"Animals activate DBL-1 pathway activity in response to Gram-negative bacteria and strongly repress it in response to select Gram-positive bacteria, demonstrating bacteria-responsive regulation of DBL-1 signaling."

Minor comments:Page 6: "The reduced feeding in response to select bacteria correlates with lifespan extension observed in both wild-type and dbl-1(-) populations." If you look at the survival curves in Figure 1 the lifespan extension of dbl-1(-) on some bacterial strains is hard to detect. This statement would be easier to understand with a reference to Table S3.

We added an in-text citation for Table S3 (and Figure 2) to this sentence.

Page 8: "We next asked if the different avoidance responses to Gram-negative and Gram- positive bacteria are associated with altered gene expression of the DBL-1 Smads."If this is the question and the driver for testing gene expression, then it would be much easier for the reader to see the potential association with presenting the RT-PCR data bundled for bacterial treatment and not organized according to the tested gene. I assume that the presentation might be caused by the RT-PCR plate design and probably not possible to reorganize. Please see this as a comment for future data presentation.

We appreciate the insightful comment and will address future RT-PCR experiments with it in mind. They are correct in their assumption that we designed the RT-PCR plates a specific way and reorganizing is not possible.

I would appreciate the gene expression lists of the RNAseq experiment to understand how the selection of immune genes for the reporter strains was made. Further, I could imagine that researchers might be interested in looking for their genes of interest in the dbl-1 mutant and/or bacterial infection context. Is the raw data uploaded to the standard sequencing data repositories?

We hope that our gene expression lists will be useful for our colleagues to data mine! Data will be uploaded to repositories when needed for this manuscript process.

Discussion: Another trigger for the worms' avoidance response could be intestinal distention by bacterial accumulation as it was described in a recent paper for *E. faecalis* (DOI: https://doi.org/10.7554/eLife.65935), and also associated with DAF-7/TGF-β signaling in the context of PA14 avoidance (doi: 10.7554/eLife.50033).

We have used these two references to expand on our discussion about possible mechanisms for how DBL-1 affects behavior, paragraph 3 (new/revised wording is bolded here and throughout this response):

DAF-7 also contributes to the perception of and avoidance of *P. aeruginosa* (Meisel et al., 2014, Singh and Aballay, 2019). Future work will determine if DBL-1 affects avoidance behavior by repressing DAF-7 signaling. Another possibility is that DBL-1 signaling responds to physiological changes caused by bacteria; distension of the intestinal lumen by bacterial colonization has been shown to elicit avoidance behaviors (Singh and Aballay, 2019, Filipowicz et al., 2021).

Page 17: The formula for calculating CFU/worm does not take the number of worms into account.

In the Materials and methods, we changed the formula to include the five animals:

“Bacterial colony forming units (CFU) per worm was calculated using the formula: (colony number × dilution factor)/ (5 animals × 100 μl lysate plated).

Figure 2, Figure 5, Figure 6H, Figure 7: Whenever there is a repeated comparison to the OP50 control (and an unpaired t-test was performed) a correction for multiple testing is missing which might alter some of the significant values.

This reviewer’s comment allowed us to (1) better note the statistics we did use for Figures 2 and 7, which included ANOVA, and (2) apply ANOVA to the data presented in Figures 5 and 6H. ANOVA includes a correction factor.

For the data presented in Figures 5 and 6H, we performed one-way ANOVA to identify overall significant differences within groups, followed by two-tailed unpaired *t*-tests to identify significant differences between specific groups at a 95% confidence interval.

In the legends to Figure 5 and 6, we changed the statistical analysis description to read:

* *p* <0.05 … by one-way ANOVA followed by unpaired *t*-test.

We revised the Materials and methods Statistical Analysis paragraph: The average pharyngeal pumping rates were analyzed using two-way ANOVA and performed pair-wise comparisons using the two-tailed unpaired *t*-test. Bacterial colonization counts (CFU per worm) were compared using the two-tailed unpaired *t*-test. The intestinal barrier function phenotypes were statistically analyzed by the Chi-square test. The avoidance ratio was compared by repeated measures ANOVA using Tukey’s post-hoc test. qRT-PCR values and mean fluorescence intensities were evaluated using ANOVA followed by a two-tailed unpaired *t*-test.

Figure 3 and 4: What does n mean in these assays? 30 worms per plate with a total of three plates (as stated in the Material and Methods)? This would be actually n = 3 and not n = 30.

We clarified the figure legends by removing “n=30 per condition” and replacing it with the more precisely worded sentence, “Each trial used three plates of 30 animals each per condition.”

Figure 5, Figure S2: Could you please describe the triplicated experiment for the RT-PCR as you did in the Material and Methods section (3 technical replicates, 3 independent biological runs)? That would be easier to understand.

We expanded the methods descriptions in the legends to Figure 5 and S2 to read: “Experiments were performed in three technical replicates and in three independent biological trials”.

Figure S1 and Material and Methods: The classification of leakiness in a number code (1-3) is not necessary for understanding the result. You could omit this to avoid confusion.

We omitted the number code. We agree that this simplifies the presentation of this work.

Figure S3: Looking at the RNAseq heatmaps it is not clear why these reporter strains have been chosen in the first place. dod-22 and dod-24, for example, do not seem to show a significant difference in WT and dbl-1(-) (heatmap S. marcescens Figure S3E). Additionally, it would be also interesting to find the selected genes of the reporters in all of the heatmaps (one of the genes is present in heatmap Figure S3F).

These genes were selected because of (1) previously published responsiveness to Gram-negative, Gram-positive, or both types of bacteria, and (2) their change in response to *S. marcescens*, *E. faecalis*, or both, either compared to the *E. coli* control or (3) their change in wild-type compared to the *dbl-1* mutant. Genes that were not differently expressed in animals lacking DBL-1 would not be included in the heatmaps. We have justifications for the selected genes in the text. Here is a table that displays the permutations these genes represent:

**Author response table 1. sa2table1:** 

gene	Published change onGram-negative bacteria	Published change on Gram-positive bacteria	Published target of IIR pathway	Change with *dbl-1(-)* compared to wild typeon *E. coli* OP50 (thiswork)	Change with *dbl-1(-)* compared to wild typeon *S. marcescens* (thiswork)	Change with *dbl-1(-)* compared to wild typeon *E. faecalis* (our work)	Change with wild-type on *S. marcescens*compared to wild typeon *E. coli* OP50 (ourwork)	Change with wild-type on *E. faecalis* compared to wild typeon *E. coli* OP50 (ourwork)
*dod-22*	Y	N	DAF-16	N	N	Y	Y	Y
*F55G11.7*	Y	Y	DAF-16, MAPK, and DBL-1	N	Y	N	Y	N
*irg-4*	Y	N	DAF-16, MAPK, and DBL-1	Y	Y	Y	Y	Y
*dod-24*	Y	N	DAF-16	N	N	Y	Y	Y
*ilys-3*	N	Y	ERK MAPK	N	N	N	N	Y

Page 12: "…but was wild type in response to *E. faecalis*, consistent with our RNA-seq results" or "and validate some of the RNA-sequencing results."These statements are difficult to reconstruct if ilys-3 and the other reporter genes' expression is not shown in Figure S3 (or discussed anywhere else).

Because *ilys-3* is unchanged between the wild type and *dbl-1* mutant strain, gene expression changes are not noted in Figure S3 (the change in *ilys-3* gene expression is between *E. coli* control and *E. faecalis*). The data in question are in the dataset that will be deposited in an online repository when needed for this manuscript process.

Table S4: I generally would prefer to look at "positive" results, i.e. "% of animals WITH detectable fluorescence" (and not "with NO detectable fluorescence").

The “positive results”—or changed animals—are really the animals lacking fluorescence. We rephrased the table heading to “% animals lacking detectable fluorescence”.

Data S1: The link does not work anymore. Maybe instead of a link you could provide the list of genes you used to generate the sunburst graphs with WormCat.

We have new links to the sunburst link/static sunburst plot. Data used to generate these plots will be uploaded to repositories when needed for this manuscript process.

Reviewer #2 (Evidence, reproducibility and clarity (Required)):Major and minor comments:From the text it appears that one representative experiment is shown. Why not include data from all experiments in supplemental table with lifespan data?

We have included data from trials not represented in Figure 1 in Table S3.

The links to data in S1 on WormCat enrichment are broken.

See response to Reviewer #1.

The use of FuDR and its possible influence on the results should be discussed.

We discussed the possible effect of FUdR on survival analyses in the Results section. While FUdR can affect the lifespan of some mutant animals, previous work from Mallo et al. (2002) suggested that FUdR does not significantly affect the lifespan of *E. coli*-fed animals lacking DBL-1. We confirmed that in our work before testing other bacterial conditions (Figure 1). We addressed the possible effect of FUdR in the results. We grew animals on plates containing 5-fluorodeoxyuridine (FUdR). While FUdR extends lifespan of wild-type animals, we used it to prevent offspring production so we wouldn’t need to transfer animals to fresh plates each day, thereby minimizing possible damage (and possible early death) to the tested animals. Our test and control are on FUdR plates, therefore the comparisons are not affected by presence of this common variable.

Is the mechanism of virulence known for the different bacteria? Do they secrete toxins, do they need to be alive etc. This should be discussed in terms of the avoidance behavior.

While a *S. marcescens* virulence factor, serrawettin W2, is mentioned in the Discussion, the other bacterial virulence mechanisms and how *C. elegans* responds to them are fascinating questions that prompted us to write in this manuscript, “It will be of interest to determine the bacterial cues that are responsible for eliciting DBL-1-mediated avoidance behavior.”

The consequences of avoidance in the laboratory setting, where alternatives in terms of food are not present for the worms, should be discussed. I suppose avoidance could lead to DR – and thus a positive effect – or it could cause starvation which in the longer term would be detrimental. Can the underlying mechanism be resolved from the current data.

We recognize that avoidance can affect survival in a complex way, but also recognize that our results are insufficient to make conclusions regarding an underlying mechanism. This is an excellent question for future work, for which this work had laid a solid foundation.

Could representative pictures of a mild and severe Smurf scored worm be included?

We have included representative images of how we scored no, mild, and severe Smurf phenotypes in Figure S1.

Figure 1: headline is kind of misleading – OP50 is an exception?

We changed the figure heading to read, “Loss of DBL-1 decreases lifespan of animals exposed to test Gram-negative and Gram-positive bacteria”.

P12: (Figure 7J) should be move forward in the sentence – the data for ilys-3 is not shown in the RNA-seq results?

We moved “(Figure 7J)” from the end of the sentence to support the results it shows: “However, *ilys-3* reporter activity also remained at relatively low levels upon loss of DBL-1 in response to Gram-positive bacteria *B. megaterium* and *S. epidermidis*, but was wild type in response to *E. faecalis* (Figure 7J), consistent with our RNA-seq results.”

Reviewer #3 (Evidence, reproducibility and clarity (Required)):– Table S3 shows the effect of dbl-1 on life span and the effect of each pathogen on wild type or dbl-1. Figure 1 shows only a part of the results reported in Table S3. It will be more clear to show the complete set of the results listed in Table S3 in Figure 1 or present Table S3 as a main Table.

See response to Reviewer #2.

– Their findings on sma-4 in regulating the avoidance of Gram-positive bacteria suggest that sma-4 acts independently of dbl-1. However, this new functional understanding of sma-4 is based on a single sma-4 mutant. It will be more convincing to test another allele of sma-4 or RNAi of sma-4.

While we agree that having another allele of *sma-4* recapitulate the phenotypes observed with *sma-4(e729)*, a known null mutation, would help validate our conclusions, the results we generated by behavioral (avoidance) and molecular (real-time PCR) assays support our conclusion that SMA-4 is acting partly independently of the DBL-1 signaling pathway in innate immune responses. Because of the evidence presented in this work that SMA-4/co-Smad is acting independently of the DBL-1 signaling pathway in some conditions, we plan to expand on this in future work. We have data of LW5556 *sma-4(jj278)* (another allele of *sma-4*) avoidance on OP50 that supports our avoidance results with *sma-4(e729)* (three biological replicates, A=0.87 at 48 hours).

– In the discussion, for "DBL-1 is known to be secreted from the AVA interneurons and activates DBL-1 pathway signaling in the hypodermis to regulate aversive learning upon exposure to Gram-negative *P. aeruginosa*. However, the neuronal circuit(s) used by DBL-1 to direct aversive behaviors remains to be identified (Zhang and Zhang, 2012)", Olofsson, B. 2014 should also be cited here and the study shows that the pathway of DBL-1 for avoiding bacteria is different from the DBL-1 pathway for learning.

We added a sentence to address the findings of Olofsson, 2014, in the discussion paragraph in question: “DBL-1 is required for inhibiting avoidance behavior and for aversive learning, but the cells involved in both DBL-1 secretion and reception are different for these two behaviors (Zhang and Zhang, 2012, Olofsson, 2014).” We think this inclusion further enhances the discussion and thank the reviewer for this recommendation.

– The links for Data S1 do not work.

See response to Reviewer #1.

– The authors use EZ media plates to maintain worms in this study. EZ media plates appear different from NGM plates that are used in most studies on *C. elegans*. Do worms maintained on EZ plates show any difference in life span, innate immunes responses or other assays used in the study compared with worms maintained on NGM?

The mean lifespan wild-type animals on EZ plates or NGM plates is similar (our unpublished observations). We compared the expression of a DBL-1 pathway reporter in animals grown on NGM plates (Roberts et al., 2010) to animals grown on EZ plates (Madhu et al., 2020) and they are qualitatively comparable. We have not directly compared innate immune responses of animals grown on NGM vs. EZ plates. All animals were grown on the same media so the media is a constant in these experiments, not a variable.

– In the intestinal barrier function assay, how do the authors quantify the leakage/Smurf as mild or severe?

See response to Reviewer #2.

– The authors need to describe the temperature at which the microbial avoidance assays were performed.

We wrote in the Strains and Maintenance section of Materials and methods, “All *C. elegans* strains were maintained on EZ media plates at 20°C”. This includes all the assays, including the avoidance assays.

[Editors' note: further revisions were suggested prior to acceptance, as described below.]

Based on the reviewers' comments, we are potentially interested in this work. However, the editors agree that the proposed revision plan is inadequate as presented. Moreover, the editors raised a few additional issues that will also need to be addressed with new data and/or experiments for reconsideration at eLife.Specifically:1. (Issue also raised by Reviewer 1) Figure 3: While it is appreciated that the RT-PCR experimental design quantified by gene and not by bacteria, it is nevertheless important that a subset of these data be repeated by bacterial treatment.

We created Figure 4—figure supplement 1 to compare these data by bacterial treatment.

2. Related to the above in terms of data organization and presentation: An important conclusion of the work is that SMA-4 acts independently of the canonical DBL-1 signaling pathway. Avoidance responses are compared in dbl-1(0) and sma mutants across time in Figures 2 and 3. However, given the variability of these assays it is difficult to compare across independent experiments. sma-4 and dbl-1 mutants should be tested in parallel in the same experiments and the data plotted in the same graph.

We performed the experiment again and created a new Figure 3 to address this and concern #3. Data from two strains with different alleles of *sma-4* (*sma-4(jj278)* was added to address this concern) are plotted in the same graph. Experiments were performed for all strains in the same experiment in triplicate, with at least three trials, n≥30 for each replicate.

3. (Issue also raised by Reviewer 3) – given the importance of the sma-4 data, data using another independent sma-4 allele need to be presented in the main figures.

We added *sma-4(jj278)* when we redid the experiment and presented results in the main figure for avoidance behavior. This *sma-4* allele is a 3,556bp deletion generated by the Jun Kelly Liu lab by CRISPR/Cas9 editing that deletes almost the entire coding region of *sma-4* and is a null *sma-4* mutation (McKillop et al., MicroPublication Biology, 2018). The *sma-4(e729)* allele used in the original submission and the resubmission is an early termination mutation, a single nucleotide substitution resulting in a Q246 (CAA) to stop (UAA) mutation (Wang et al., Development, 2002). Both *sma-4* alleles resulted in similar organismal avoidance responses.

4. (Issues raised by all reviewers) – In order for the reviewers and editors to review the data, all RNASeq data need to be included in the manuscript for review. The data should also be appropriately analyzed and should include standard and complete information including gene names, read counts, and other statistics. In the absence of the actual data, the RNASeq analyses (including the sunburst plot) cannot be appropriately reviewed, and the justification for focusing on specific genes cannot be assessed.

We have submitted source data for all figures, not just the RNASeq data.

5. (Issue also raised by Reviewer 1) – In several figures (including Figure 5 and Figure 6) – the statistics should include a correction for multiple comparisons.

We included a correction for multiple comparisons using (1) Dunnett’s multiple comparisons test for one-way ANOVA analyses for Figures 4 and 5, and (2) Tukey’s post-hoc test for two-way ANOVA analyses for Figures 2 and 6 and included the corrections in the relevant figure legends and the Statistical Analyses section.